# Transductive Visual Programming: Evolving Tool Libraries from Experience for Spatial Reasoning

**Shengguang Wu, Xiaohan Wang, Yuhui Zhang, Hao Zhu, Serena Yeung-Levy**
Stanford University

## Abstract

Spatial reasoning in 3D scenes requires precise geometric calculations that challenge vision-language models. Visual programming addresses this by decomposing problems into steps calling specialized tools, yet existing methods rely on either fixed toolsets or speculative tool induction before solving problems, resulting in suboptimal programs and poor utilization of induced tools. We present **T**ransductive **V**isual **P**rogramming (TVP), a novel framework that builds new tools from its own experience rather than speculation. TVP first solves problems using basic tools while accumulating experiential solutions into an Example Library, then abstracts recurring patterns from these programs into reusable higher-level tools for an evolving Tool Library. This allows TVP to tackle new problems with increasingly powerful tools learned from experience. On Omni3D-Bench, TVP achieves state-of-the-art performance, outperforming GPT-4o by 22% and the previous best visual programming system by 11%. Our transductively learned tools are used $5\times$ more frequently as core program dependency than inductively created ones, demonstrating more effective tool discovery and reuse. The evolved tools also show strong generalization to unseen spatial tasks, achieving superior performance on benchmarks from SpatialScore-Hard collection without any testset-specific modification. Our work establishes experience-driven transductive tool creation as a powerful paradigm for building self-evolving visual programming agents that effectively tackle challenging spatial reasoning tasks. We release our code at https://transductive-visualprogram.github.io/.

## 1 Introduction

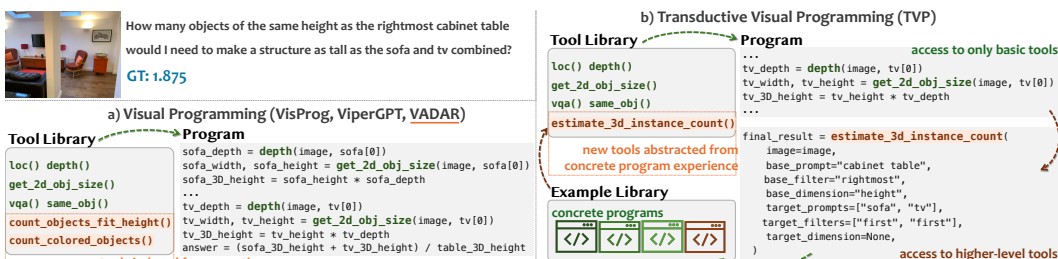

Figure 1: (a) Prior methods operate in an open-loop manner: tools are created without experience from solving problems. (b) TVP maintains both an Example Library of experiential solutions and a Tool Library of abstracted functions, forming a closed-loop system where tools are created from proven solution experience, and are then reused to guide future problem-solving.

Reasoning about 3D spatial relations in real-world scenes requires precise geometric calculations beyond visual perception alone. Despite the progress of pretrained vision-language models (VLMs) on visual question answering, precise 3D spatial reasoning with real-world dimensions remains challenging even for leading models (Lee et al., 2025; Marsili et al., 2025). Consider the query in Fig. 1: *determining how many cabinets would equal the combined height of a sofa and TV.*

The answer requires object localization, depth-aware height estimation for multiple objects, and computing the precise ratio of $1.875$—a level of precision that VLMs struggle to achieve.

This limitation motivates **visual programming**, a compositional approach that decomposes complex visual reasoning into discrete computational steps. These steps call specialized tools (*e.g.*, depth estimators, object detectors, geometric functions) and combine their outputs through programmatic logic, reaching the precision that monolithic VLMs lack.

The effectiveness of a visual programming system depends critically on **how its tool repertoire is developed**. An ideal approach would mirror how human programmers develop expertise: solving concrete problems with basic tools, recognizing recurring patterns, then abstracting them into reusable higher-level functions that make future programs simpler and more reliable.

Existing work deviates from this ideal by skipping or inverting the **experience-to-abstraction** cycle. Earlier approaches skip this cycle entirely, relying on fixed, predefined tool-sets that cannot adapt to new tasks (Gupta & Kembhavi, 2023; Surís et al., 2023). More recent work inverts the order: creating new tools through **induction**—speculating about useful functions based solely on problem descriptions—before solving them (Marsili et al., 2025). This inversion creates an open-loop system (Fig. 1, panel a): tools flow in one direction from creation to application, with no problem-solving experience to guide abstraction. Tool development requires a closed-loop process (Fig. 1, panel b) where experience with concrete programmatic solutions guides abstraction, ensuring that new tools address proven needs. Without this **experiential grounding**, speculatively induced tools often seem helpful in theory but fail to simplify problems in practice, as evidenced by severe under-utilization: $94.2\%$ of programs still rely on basic predefined tools despite having access to newly created functions (Fig. 2).

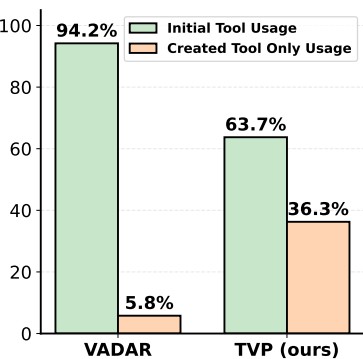

Figure 2: Tool usage distribution: transductive (TVP) vs. inductive (VADAR) abstraction.

We propose **Transductive** Visual Programming (TVP), which realizes this experience-driven ideal through a closed-loop process enabled by dual libraries. An Example Library stores successful program solutions as experiential memory, while a Tool Library contains available functions, starting from basic vision tools and expanding with abstracted tools. When solving a visual reasoning problem, TVP retrieves similar examples from the Example Library as few-shot demonstrations, then generates program candidates using functions from the Tool Library. High-quality programs are stored in the Example Library, while similar programmatic solutions are clustered and abstracted into new parameterized functions for the Tool Library. This creates a circular **program-tool-program cycle**: program solutions inform tool development, and these tools then enable better future programs (Fig. 1, panel b).

On Omni3D-Bench (Marsili et al., 2025) for 3D spatial reasoning, TVP achieves state-of-the-art performance, outperforming GPT-4o by 22% and the previous best visual programming system by 11% (§3.1). This advantage is most pronounced on the hardest batch of problems, where TVP demonstrates most drastic improvement as its dual libraries grow (Figs. 6 and 7). Compared to inductive tool creation, TVP's learned tools are used $5\times$ more frequently as core program dependency—36.3% of programs require **only** abstracted tool calls (Fig. 2). Programs using TVP's abstractions achieve both **greater accuracy** ($+3.4\%$) and **lower complexity** (66% reduction in cyclomatic complexity) (Fig. 5). The dual libraries that TVP evolves also **transfer to unseen spatial reasoning tasks**. With the libraries built by running on Omni3D-Bench, TVP delivers superior zero-shot performance on novel queries from other spatial benchmarks including 3DSR-Bench (Ma et al., 2024), SpatialSense (Yang et al., 2019), and VG-Bench (Wu et al., 2025), outperforming all baselines without any testset-specific modification (Tab. 2 & Fig. 9).

In summary, grounding tool creation in problem-solving experience produces highly-reusable abstractions that enable simpler, more precise programs and transfer well across tasks. TVP's transductive paradigm points toward programming agents that continuously improve through accumulated experience, building increasingly sophisticated capabilities from basic operations to complex reasoning functions, analogous to how human programmers develop expertise.

## 2 TRANSDUCTIVE VISUAL PROGRAMMING

Transductive Visual Programming (TVP) implements the closed-loop paradigm from Fig. 1(b) via two interconnected libraries: an Example Library $\mathcal{E}$ that accumulates program solutions as experience, and a Tool Library $\mathcal{T}$ that maintains functions abstracted from these programs. The dual-libraries enable the circular **program-tool-program** cycle: solving problems generates experience, experience guides tool creation, and newly created tools improve future problem-solving.

Given a dataset $\mathcal{D} = \{(I_i, q_i)\}_{i=1}^{N}$ of images and spatial questions, TVP alternates between two continuous phases. The first phase solves questions using available tools while accumulating high-quality solutions in $\mathcal{E}$; the second phase mines patterns from $\mathcal{E}$ to abstract new tools for $\mathcal{T}$. As more problems are solved, both libraries grow—expanding experience and increasingly sophisticated tools enable simpler, more precise programs. The complete workflow is formalized in Alg. 1, with key sub-processes detailed in Algs. 2 to 4. Fig. 3 illustrates this dual-library architecture and how the phases interact.

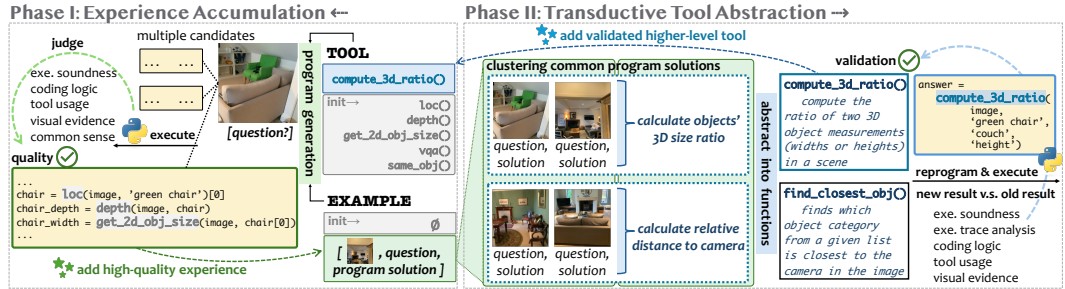

Figure 3: TVP's dual-library architecture. (Phase I) Problem-solving and experience accumulation: For each query, TVP retrieves similar examples from the Example Library and generates programs using the current Tool Library; high-quality solutions join the Example Library. (Phase II) Tool abstraction: Accumulated examples are clustered, and common patterns are abstracted into new tools, which, if passed validation, are added to the Tool Library for future use.

### 2.1 SYSTEM INITIALIZATION

TVP begins with an empty Example Library $\mathcal{E} \leftarrow \emptyset$, and a Tool Library $\mathcal{T}$ initialized with basic vision tools (inherited from Marsili et al., 2025): object localization (`loc`) and bounding box detection (`get_2d_obj_size`) with GroundingDINO (Liu et al., 2024); depth estimation (`depth`) via UniDepth (Piccinelli et al., 2024); visual question answering with GPT-4o (`vqa`); and bounding box overlap verification (`same_object`). The predefined tools provide atomic operations that compose into more complex geometric reasoning patterns.

### 2.2 PHASE I: PROBLEM-SOLVING AND EXPERIENCE ACCUMULATION

For each visual reasoning question, TVP generates programs using its current libraries. This process involves four steps: retrieving similar past solutions, generating candidate programs, executing them, and evaluating their quality. High-quality solutions enter the Example Library, building concrete problem-solving experience.

**Problem Solving Workflow. (1) Example retrieval:** Given a question $q_i$, TVP retrieves up to $k_{max}$ previously solved queries from $\mathcal{E}$ with text embedding similarity above threshold $\tau_{sim}$, as sufficiently similar questions may well require similar solution logic. Retrieved examples serve as in-context demonstrations for the program generator. If no examples meet the similarity threshold, TVP generates programs without demonstrations. **(2) Program generation:** The program generator receives the question $q_i$, the 0-$k_{max}$ retrieved examples with their solutions, and the current Tool Library $\mathcal{T}$ (tool signatures and docstrings). It samples $m$ candidate programs, because visual reasoning problems often have multiple valid approaches, and sampling increases the chance of discovering a high-quality solution. **(3) Program execution:** TVP executes each candidate program $p_j$ in Python with access to the image $I_i$ and the full Tool Library $\mathcal{T}$ implementations. Execution produces both

a final answer and a complete trace capturing all intermediate computations. Programs that fail to execute or produce no result are filtered out, leaving only valid candidates for quality evaluation. **(4) Quality judgment:** A VLM judge assesses each candidate program's quality. The judge has access to the program implementation, execution trace and the produced answer, evaluating them against the question and image as visual evidence. More details on the judge criteria are discussed in §B.3. The highest-scoring candidate becomes the final solution; if its score exceeds the quality threshold $\tau_q$, it **enters the Example Library** $\mathcal{E}$.

**Experience Accumulation and Update.** Across $T$ iterations over the entire dataset, newly added programs can replace prior solutions for the same question in $\mathcal{E}$ if they achieve higher quality scores. This replacement happens naturally as TVP abstracts more powerful tools that enable simpler, more efficient programs for previously solved questions (discussed in §2.4). The accumulated high-quality solutions in $\mathcal{E}$ form the concrete experience from which tools are abstracted.

## 2.3 PHASE II: TRANSDUCTIVE TOOL ABSTRACTION

At regular intervals ($n_a$ questions processed), TVP analyzes the Example Library to identify recurring solution patterns and abstract them into reusable tools. This abstraction is transductive: it **converts concrete program solutions directly into parameterized functions**, so that every tool is grounded in actual problem-solving experience. The abstraction process involves clustering similar solved problems, creating parameterized functions that capture shared logic, and validating these functions before adding them to $\mathcal{T}$.

**Pattern Mining Through Clustering.** TVP first clusters all queries in the current $\mathcal{E}$ by embedding similarity. For clusters where similarity exceeds $\tau_{\text{sim}}$ and size exceeds $\tau_{\text{cluster}}$, an LLM analyzes the cluster's program solutions to assess **abstraction potential**. This analysis evaluates whether the cluster shows recurring computational patterns—such as repeated *ratio calculations* or *relative camera distance* (Fig. 3)—that warrant parameterization. More details on the criteria for abstraction potential are discussed in §B.3. Clusters scoring above $\tau_{\text{potential}}$ trigger tool creation.

**Creating Parameterized Tools.** A tool abstractor receives all questions, program solutions, and execution results from the cluster, along with the current Tool Library $\mathcal{T}$. It creates a parameterized function capturing the cluster's shared computational pattern. As shown in Fig. 3, clusters repeatedly *calculating 3D size ratios* lead TVP to create `compute_3d_ratio`, while clusters *finding objects closest to camera* yield `find_closest_obj`. These parameterized tools replace multi-step programming logic with single function calls, thus simplifying solutions to similar future problems.

**Rigorous Tool Validation.** Before entering TVP's toolbox $\mathcal{T}$, each abstracted tool must pass validation to ensure it executes correctly and maintains solution quality. Validation proceeds in two stages (detailed in Alg. 3): execution validation and correctness validation. In Stage 1, TVP rewrites each program in the cluster to use the new tool, then runs the rewritten version. All examples must **execute successfully** (100% success rate)—any failure immediately rejects the proposed tool. In Stage 2, when rewritten programs produce different results from the original, a VLM judge evaluates whether the new result is **equally valid or superior** given the visual evidence. Such divergence is common for geometric calculations involving floating-point arithmetic, where multiple answers may be equally valid within numeric precision. The tool is accepted only if overall correctness (considering both identical and validated-divergent results) exceeds $\tau_{\text{correct}}$. Successfully validated tools join $\mathcal{T}$, and the cluster examples in $\mathcal{E}$ are rewritten to use the new tool.

## 2.4 CLOSING THE LOOP: TOOLS IMPROVE PROBLEM-SOLVING

With new tools added to $\mathcal{T}$, future problems in Phase I gain access to increasingly powerful abstract functions. The program generator can thus produce solutions that are simpler (lower code complexity) and more precise (better accuracy) (evidenced in Fig. 5 and discussed in §3.1). While these cleaner, more accurate programs accumulate in $\mathcal{E}$, subsequent abstraction (Phase II) work from higher-quality patterns, enabling even more sophisticated tool creation. As such, this program-tool-program cycle continuously improves TVP's dual-libraries.

## 2.5 TOOL LIBRARY MAINTENANCE

As TVP processes more questions and performs abstraction at regular intervals, functionally similar tools may emerge from different clusters. Therefore, at every $n_d$ questions, TVP **identifies and merges similar tools** (detailed in Alg. 4 and illustrated in Fig. 4). For instance, tools like `compute_3d_ratio` and `compute_3d_group_size_ratio` merge into a more general `compute_objects_size_ratio` that handles both *pairwise* and *group* ratio calculations. Merged tools undergo the same two-stage validation (§2.3) against all examples that used any of the original tools, ensuring no functionality is lost. This maintenance keeps the Tool Library concise and organized, making it easier for the program generator to select appropriate tools.

Through repeated merging, tools evolve into **increasingly general** abstractions that cover broader functionality, as traced in Fig. 11 and discussed in §3.3; Fig. 10(a) shows concrete program updates via these evolved tools that become progressively simpler across iterations.

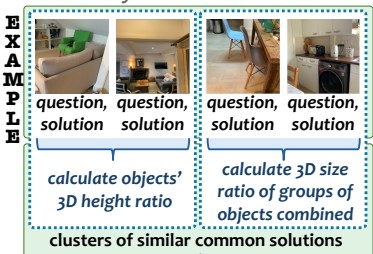

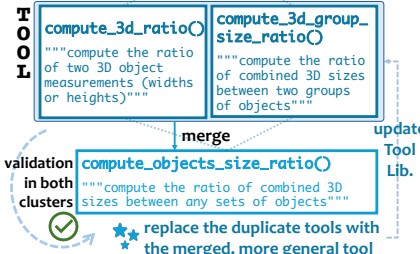

Figure 4: Tool Library maintenance via merging (§2.5). Functionally similar tools emerge from different clusters; TVP merges them into a more general abstraction covering both use cases.

## 3 EXPERIMENTS

We test TVP on 3D spatial reasoning, a domain requiring geometric precision that significantly challenge monolithic VLMs (§1). Our experiments demonstrate three key results: (1) TVP achieves superior performance through transductive tool creation—with especially strong advantages on hard questions; (2) TVP's dual libraries continuously evolve to produce increasingly efficient and accurate programs; (3) the learned dual libraries generalize strongly to unseen spatial reasoning benchmarks.

### 3.1 3D SPATIAL REASONING PERFORMANCE

**Setup.** We evaluate on Omni3D-Bench (Marsili et al., 2025), a challenging test set of 501 non-templated spatial queries on real-world scenes requiring 3D geometric reasoning. The benchmark includes Yes/No, Multiple Choice, and Counting questions (evaluated by exact-match accuracy after minor string normalization); and Floating-point calculations evaluated via Mean Relative Accuracy (Yang et al., 2025a): $\mathcal{MRA} = \frac{1}{|C|}\sum_{\theta \in C} \mathbb{1}\left(\frac{|\hat{y}-y|}{y} < 1-\theta\right)$ with $C = \{0.5, 0.55, ..., 0.95\}$, and Float($\pm$10%) accuracy for predictions within 10% tolerance. Overall accuracy counts correct answers across all question types, using Float($\pm$10%) for floating-point questions. We compare against three baseline categories: **generic VLMs** (GPT-4o, LLaVA-OneVision-7B-Chat Li et al., 2024, Qwen2-VL-7B-Instruct Wang et al., 2024a, Molmo-7B-D Deitke et al., 2025), **spatial-finetuned VLMs** (SpaceMantis Jiang et al., 2024[1], SpatialBot-3B Cai et al., 2025), and prior **visual programming systems** (VisProg Gupta & Kembhavi, 2023, ViperGPT Surís et al., 2023, VADAR Marsili et al., 2025). We run TVP for $T = 3$ iterations using GPT-4o for program generation and quality judgment, with tool abstraction and merging at every step ($n_a = n_d = 1$). Additional implementation details are in §B.

**TVP achieves state-of-the-art through experience-grounded tool creation.** TVP scores 33.3% overall accuracy on Omni3D-Bench (Tab. 1), outperforming the previous best visual programming system VADAR (29.9%) by +11% and GPT-4o (27.2%) by +22%. Notably, **monolithic VLMs** handle perception-heavy tasks reasonably well (GPT-4o reaches 65.3% on yes/no questions) but **struggle with exact 3D measurements** requiring multi-step geometric reasoning and arithmetic operations (8.2% on float calculations). Even spatial-finetuned models like SpaceMantis show no improvement over generic VLMs, again reflecting the necessity of compositional approach for precise spatial reasoning.

---

[1]Finetuned following SpatialVLM (Chen et al., 2024)

Table 1: Performance on Omni3D-Bench. TVP achieves state-of-the-art through transductive tool learning. **Best bold**, second underlined. *Results from VADAR paper.

| Method | Accuracy by Question Type (%) | | | | | Overall (%) |
|---|---|---|---|---|---|---|
| | Yes/No | Multiple Choice | Counting | Float MRA | Float ($\pm$10%) | |
| *Generic VLMs* | | | | | | |
| GPT-4o | **65.3** | 60.5 | 18.6 | 26.7 | 8.2 | 27.2 |
| Qwen2-VL-7B-Inst | 58.7 | 33.7 | 12.9 | 21.5 | 10.0 | 21.8 |
| LLaVA-OV-7B-Chat | 60.0 | 27.9 | 22.9 | 26.8 | 11.1 | 23.0 |
| Molmo-7B-D | 46.7 | 41.9 | 18.6 | 28.4 | 8.9 | 21.6 |
| *Spatial-Finetuned VLMs* | | | | | | |
| SpaceMantis | 53.3 | 30.2 | 4.3 | 21.4 | 8.2 | 18.2 |
| SpatialBot-3B | 60.0 | 30.2 | 0.0 | 17.7 | 8.5 | 18.8 |
| *Visual Programming* | | | | | | |
| VisProg* | 54.7 | 25.9 | 2.9 | 0.9 | — | — |
| ViperGPT* | 56.0 | 42.4 | 20.0 | 15.4 | — | — |
| VADAR | 56.0* | 57.6* | 21.7* | 35.5* | 15.9 | 29.9 |
| **TVP (ours)** | 60.0 | **61.6** | **24.3** | **36.5** | **19.3** | **33.3** |
| TVP w/o Tool Lib | 60.0 | **61.6** | 21.4 | 35.5 | 17.0 | 31.7 |

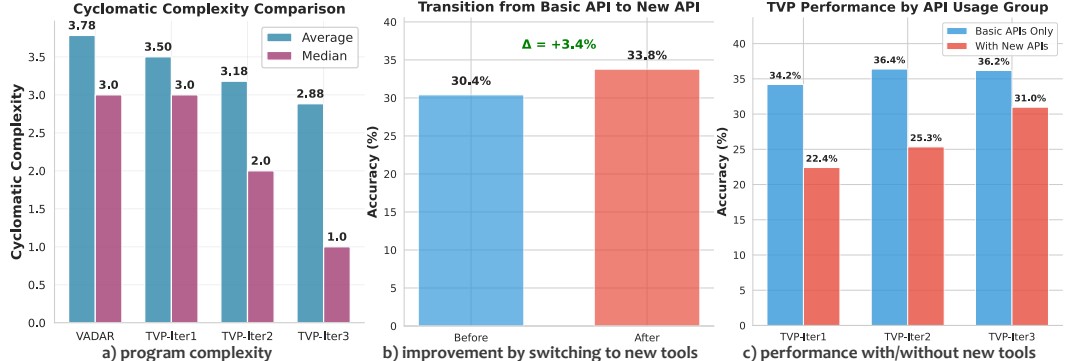

Figure 5: TVP's closed-loop learning produces three measurable benefits. (a) Program complexity decreases as higher-level tools replace multi-step patterns. (b) Programs gain +3.4% accuracy when switching to abstracted tools. (c) Performance with abstracted tools improves +38% across iterations as TVP learns more capable tools and better examples.

**TVP enables simpler, more accurate, and continuously improving programs.** Fig. 5 reveals three facets of how transductive tool abstraction improves programs. First, **program complexity steadily decreases (panel a)**, with median McCabe's Cyclomatic Complexity Number (CCN) (Mc-Cabe, 1976) (§B.2) dropping from 3.0 to 1.0 across iterations. This reduction occurs because abstracted tools replace multi-step programming patterns with single parameterized function calls. Second, **programs gain +3.4% accuracy when switching to abstracted tools (panel b)**. This improvement stems from TVP's experience-grounded tool creation: each abstracted function encapsulates recurring high-quality solutions and passes rigorous validation, thus reducing potential error rates compared to reimplementation with basic tools. Third, **programs utilizing abstracted tools improve significantly across iterations (panel c)**—from 22.4% accuracy in iteration 1 to 31.0% in iteration 3, a +38% relative gain—while programs using only basic tools maintain stable performance. This improvement reflects the closed-loop refinement process (§2.4): later iterations benefit from (1) more sophisticated tools abstracted, and (2) higher-quality examples demonstrating optimal tool usage patterns.

**Tool abstraction facilitates progressive self-improvement, especially on hard problems.** To isolate the Tool Library's contribution beyond in-context learning from examples, we run TVP with

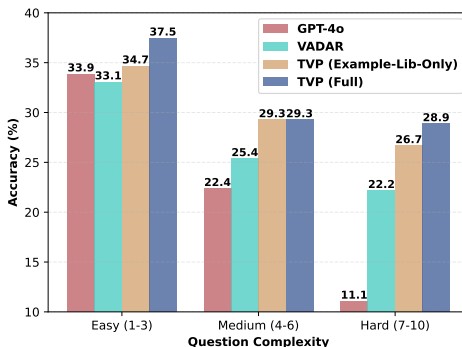

Figure 6: Performance comparison across question complexity levels on Omni3D-Bench. The complexity scores are rated with criteria defined in §B.4

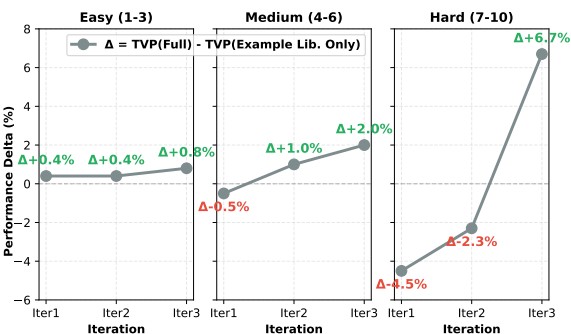

Figure 7: Performance delta between TVP (Full) and TVP (Example-Lib-Only) across iterations for each question complexity level. TVP full system (with active tool creation) shows most significant gains on the hardest batch of questions.

only the Example Library active ("w/o Tool Lib" in Tab. 1), providing retrieved examples as few-shot demonstrations but restricting the program generator to the 5 basic initial tools from §2.1. Notably, **the Example Library alone maintains competitive performance** (31.7% overall, outperforming all prior baselines). This strong performance demonstrates the quality of our accumulated experience, enabled by TVP's example library admission design.

**The full TVP system with active tool creation shows more significant improvement across iterations** (31.3% → 31.9% → **33.3**%), while the Example-Library-Only variant maintains static (**31.7**% → 31.5% → 31.5%). The progressive improvement stems from the **closed-loop design** in Fig. 1(b): abstracted tools encapsulate past experience and enable better future programs, which become better examples, from which better tools can be abstracted. Fig. 5(b) also indicates this effect: when programs switch from basic tools to newly created abstractions, they achieve +3.4% accuracy improvement. Without tool creation in the loop, this self-improving cycle is weakened.

Furthermore, we find that **the tool abstraction contributes most value on hard problems**. We use GPT-5 to rate the difficulty of the spatial reasoning questions on a 1.0–10.0 scale (details in §B.4), then divide questions into three groups: Easy (1–3), Medium (4–6), and Hard (7–10). Fig. 6 shows accuracy across methods for different complexity levels. **TVP (Full) delivers the best performance on both Easy and Hard batches**. For easy questions, thoroughly validated created tools avoid potential reimplementation errors, leading to more stable performance. For harder questions, created tools provide simpler solution steps that eliminate complicated logic, thus easing the program reasoning. Fig. 7 reveals the evolution of the benefits brought by our active tool abstraction across iterations, as we compare the the performance delta between TVP (Full) and TVP (Example-Lib-Only) for each complexity level. **On the hardest batch, TVP (Full) shows the most significant improvement trajectory**, starting at −4.5% relative to Example-Lib-Only in iteration 1, but ultimately surpassing it by +6.7% in iteration 3. This demonstrates that our created tools—beyond just in-context examples—effectively reduce the reasoning workload for most challenging questions, as they encapsulate past experience of complicated code logic into simple function calls. Examples are illustrated in Fig. 10, where newly created tools serve as convenient single-step solutions for otherwise complex spatial reasoning calculations.

**TVP's dual libraries evolve steadily through closed-loop interaction.** Fig. 12 visualizes how TVP's dual libraries evolve through three iterations on Omni3D-Bench, tracing the closed-loop program-tool-program cycle introduced in Fig. 1(b). The Example Library grows steadily from 0 to 304 high-quality solutions as TVP processes questions, accumulating concrete problem-solving experience that grounds tool abstraction. The Tool Library expands with controlled maintenance: while 61 tools are created total, only 11 remain active after periodic merging (§2.5). This selective retention ensures a manageable Tool Library capturing genuinely reusable abstractions with minimal redundancy, making it easier for the program generator to select appropriate tools.

**TVP is robust to backbone LLM choice, showing a clear scaling trend with model sizes.**

We further investigate TVP's robustness to open-source smaller LLMs, represented by the Qwen2.5-Coder-Instruct (Hui et al., 2024) family as the backbone program generator, spanning from 7B to 32B parameters. Specific configurations are in §B.2. Fig. 8 presents the scaling behavior, where TVP exhibits a clear performance improvement with increasing model capacity. Notably, using an open-source 32B model, TVP achieves performance close to our GPT-4o-backed variant (30.7% vs. 31.3%), and surpasses the previous best baseline VADAR (29.9%) despite its more capable GPT-4o backbone. This result underscores that **TVP does not rely on proprietary-specific optimal LLMs**, and **strong performance can be achieved with more accessible open-source alternatives**. The consistent scaling trend also validates TVP's architecture as model-agnostic, and suggests significant future potential of transductive tool creation, as foundation models with enhanced capabilities become available.

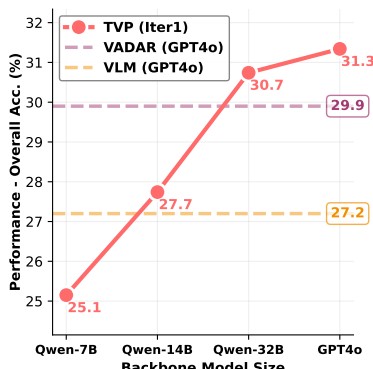

Figure 8: TVP performance scales consistently with backbone model capacity.

Additional empirical analyses on TVP's robustness to random data ordering are provided in §C.1.

## 3.2 GENERALIZING TO UNSEEN SPATIAL REASONING QUERIES

Generalization to unseen tasks provides a critical test of our transductive paradigm: tools abstracted from experience on one benchmark should capture fundamental spatial reasoning patterns that transfer to new questions. We investigate whether TVP's dual libraries, built only on Omni3D-Bench, transfer to unseen spatial reasoning queries without any modification.

**Setup.** We evaluate on SpatialScore-*Hard* collection (Wu et al., 2025), drawing 256 samples from 3DSR-Bench (Ma et al., 2024), SpatialSense (Yang et al., 2019), and VG-Bench (Wu et al., 2025) that ask single-image questions without visual bounding box hints (matching Omni3D-Bench's data structure). These queries span four spatial reasoning categories (Fig. 9): object properties, object localization, depth and distance estimation, and 3D positional relations. Yes/no and multiple-choice questions are measured with accuracy, and numeric calculations with accuracy within ±10% tolerance. We compare TVP's zero-shot transfer against the same VLM baselines from §3.1 and VADAR as the representative visual programming system, noting that VADAR creates new tools specifically for these test sets while TVP uses its Omni3D-Bench libraries zero-shot.

Table 2: Results on benchmarks from sampled SpatialScore-*Hard* collection. TVP generalizes zero-shot with only libraries built from Omni3D-Bench. **Best bold**, second underlined.

| Method | 3DSR-B. | SpatialSense | VG-B. | Overall |
|---|---|---|---|---|
| *Generic VLMs* | | | | |
| GPT-4o | 52.1 | 46.5 | 20.3 | 42.6 |
| LLaVA-OV-7B-Chat | 12.4 | 9.9 | 9.4 | 10.9 |
| Qwen2-VL-7B-Inst | 49.6 | 32.4 | 7.8 | 34.4 |
| Molmo-7B-D | 41.3 | 54.9 | 12.5 | 37.9 |
| *Spatial-Finetuned VLMs* | | | | |
| SpaceMantis | 37.2 | 19.7 | 7.8 | 25.0 |
| SpatialBot-3B | 20.7 | **62.0** | 6.2 | 28.5 |
| *Visual Programming* | | | | |
| VADAR | 24.8 | 40.8 | 39.1 | 32.8 |
| **TVP**Generalize | **52.9** | 59.2 | **43.8** | **52.3** |

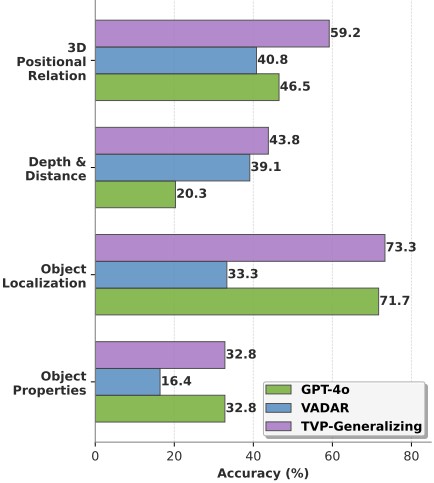

Figure 9: TVP's libraries transfer well on SpatialScore-*Hard*, particularly for 3D spatial and depth/distance reasoning.

**Transductively created tools generalize with strong zero-shot performance.** As shown in Tab. 2, TVP achieves 52.3% overall accuracy, substantially outperforming baseline VLMs and VADAR (32.8%). While VADAR's inductive approach creates specific new tools by analyzing these test set questions, TVP's experience-grounded tools still lead to more effective solutions even in zero-shot transfer. The category-wise breakdown in Fig. 9 further reveals **TVP's particular strength on challenging spatial reasoning** consistent with §3.1: 59.2% on 3D positional relations and 43.8% on depth and distance estimation.

Strong transfer occurs because **our transductive tool creation inherently produces general functions** through three mechanisms. First, tools are abstracted from clusters of similar problems (Alg. 2), forcing abstractions to capture shared computational logic rather than query-specific details. Second, rigorous validation (Alg. 3) ensures the abstraction generalizes correctly within its cluster before entering the Tool Library. Third, tool maintenance (Alg. 4) merges functionally similar abstractions, leading to more general functions validated on the union of their source examples. Together, these mechanisms ensure tools encode fundamental reasoning patterns validated on diverse examples, thus enabling effective generalization to queries of different phrasings and contexts.

### 3.3 QUALITATIVE ANALYSIS OF TOOL UTILIZATION AND EVOLUTION

**Transductively abstracted tools apply to diverse problems.** Fig. 10 illustrates TVP's hierarchically evolving tools that simplify programs and apply to diverse problems, both within Omni3D-Bench and in zero-shot transfer.

Panel (a) demonstrates how program solutions simplify as tools evolve (complementing the conceptual illustration in Fig. 11). The same spatial query—*computing the combined 3D height of a TV and TV stand given the sofa's reference height*—is solved using progressively more general tools: from basic `_compute_3d_object_height` calls, to the unified `_compute_3d_object_size` handling **multiple dimensions**, to the most general `_estimate_3d_sizes_from_reference` that **aggregates** multiple target objects in a single call. This evolution reduces program complexity while maintaining correctness.

Panel (b) shows `_compute_3d_dimension_match_count` handling diverse dimension-matching calculations within Omni3D-Bench—from *determining how many small stools fit in an armchair's volume* to calculating *how many rightmost stools must stack to reach a chair's height*. Such versatility arises from our transductive tool creation (§2.3) that abstract general underlying pattern from a cluster of examples while parameterizing specific objects, dimensions and selectors.

Panel (c) demonstrates **tool transfer across datasets**: `_find_largest_by_3d_metric`, abstracted from Omni3D-Bench, is directly applied to 3DSR-Bench queries *comparing train and street light elevations*. Successful transfer occurs because the tool captures fundamental reasoning logic ("compare multiple objects on a specified 3D metric and identify the largest") that generalizes across object categories and scene types, regardless of domain specifics.

**Tool hierarchies emerge through iterative generalization.** TVP produces increasingly sophisticated tool hierarchies through iterations of abstraction and library maintenance. Fig. 11 traces the evolution of a representative tool hierarchy. Starting from `compute_3d_height_ratio` that computes ratios of summed 3D heights between object groups, TVP generalizes this pattern via a dimension parameter, creating `_compute_group_3d_dimension_ratio` that handles both height and width calculations uniformly. As more examples accumulate, subsequent abstraction cycles reveal patterns requiring selective object filtering, leading to the merged `_compute_objects_size_ratio` with conditional filtering capabilities—an even more capable tool that subsumes the earlier versions. Fig. 10(a) shows concrete program updates with an evolving hierarchy of tools: the same spatial query is solved using progressively more general tools, with multi-step solutions collapsing into a convenient single-step function call. The quantitative impact of these tool hierarchies is also clear: programs using increasingly evolved tools achieve both higher accuracy and lower complexity (Fig. 5).

This continuous refinement demonstrates that TVP doesn't just solve problems but progressively learns more sophisticated reasoning capabilities from basic operations to increasingly powerful abstractions, mirroring how human programmers develop expertise.

## 4 RELATED WORK

### 4.1 SPATIAL REASONING

Spatial reasoning requires precise understanding of real-world 3D relationships beyond pixel-level image representations (Kamath et al., 2023; Majumdar et al., 2024; Fu et al., 2024; Tong et al., 2024; Cai et al., 2025; Zhang et al., 2024; Yang et al., 2025b), which remains challenging for monolithic VLMs. Even with specialized spatial finetuning (Chen et al., 2024; Cheng et al., 2024; Cai et al., 2025), these models still struggle with diverse 3D reasoning queries (Lee et al., 2025; Marsili et al., 2025)—consistent with our findings in §3.1. These limitations motivate visual programming approaches that decompose visual tasks into steps calling expert vision tools. Vis-Prog (Gupta & Kembhavi, 2023) uses a domain-specific language for composing vision specialists, while ViperGPT (Surís et al., 2023) generates Python code to call vision APIs; both are limited to static predefined tools. VADAR (Marsili et al., 2025) introduces a dynamic tool set but relies on pure **induction**—speculating about potentially useful functions solely from question texts before solving any problems. This leads to under-utilization of created tools in practice, as shown in Fig. 2.

TVP takes a fundamentally different approach to visual programming by learning tools **transductively** through experience: it first solves problems with basic tools, accumulates experiential solutions, and only then abstracts recurring patterns from these proven solutions into new functions.

### 4.2 TOOL USE AND ABSTRACTION

Agentic systems calling specialized tools have demonstrated strong performance across web navigation (Zheng et al., 2025; Wang et al., 2025), robotic control (Liang et al., 2023), graphics generation (Hu et al., 2024; Sun et al., 2025), game exploration (Wang et al., 2023), and biomedical experiments (Jin et al., 2025). In 3D visual tasks, Yuan et al. (2024) proposes view-dependent and -independent modules in visual programs for zero-shot open-vocabulary grounding. Mi et al. (2025) introduces "code as spatial relation encoders" optimized through test suites before deployment. Kamali & Kordjamshidi (2025) uses symbolic logic operators to enhance the expressiveness of visual programming. Beyond applying existing tools, recent work studies automatic tool creation for self-evolving agents (Cai et al., 2023; Wang et al., 2024b; Yuan et al., 2023; Qian et al., 2023; Jin et al., 2025). LILO (Grand et al., 2023) abstracts tools by compressing programs into symbolic $\lambda$-expressions. Alita (Qiu et al., 2025) creates specialized model context protocols connecting web search with tool generation and execution. Skillweaver (Zheng et al., 2025) identifies novel skills from web tasks through a simple-to-complex curriculum. ASI (Wang et al., 2025) shares our insight that tool abstractions should emerge from experience, proposing new functions from individual action trajectories in web environments.

TVP evolves its toolbox through a dual-library architecture: the Example Library accumulates concrete programs across many problems, and the Tool Library abstracts reusable functions from clusters of these examples. Their closed-loop interaction ensures that newly created tools, grounded in diverse experiences, will in turn facilitate solving a broad range of future problems.

## 5 CONCLUSION

We introduce Transductive Visual Programming (TVP), a framework that learns to build tools from concrete problem-solving experience, mirroring how human programmers develop expertise from practice. Through a program-tool-program cycle—where solving problems generates experience, experience guides tool creation, and new tools improve future solutions—TVP continuously evolves its own visual programming capabilities. We find three measurable benefits: programs simplify as recurring patterns compress into tools, accuracy increases with higher-level tool abstractions, and performance grows across iterations as tool usage improves. TVP achieves state-of-the-art on the challenging 3D spatial reasoning benchmark Omni3D-Bench, and the evolved tools transfer to unseen spatial tasks—evidence that transductive learning captures fundamental reasoning patterns beyond dataset-specific solutions. Our transductive paradigm offers a general architecture for self-evolving agents that develop increasingly sophisticated capabilities through accumulated experience—from basic operations to complex reasoning functions—across any domain requiring compositional problem-solving.

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

## A   THE USE OF LARGE LANGUAGE MODELS (LLMS)

In this paper, we used LLMs to polish the manuscript's presentation, including grammar checking, light editing, and styling of figures and tables.

## B   IMPLEMENTATION DETAILS

### B.1   VADAR REPRODUCING CONFIGURATIONS

We directly utilized VADAR (Marsili et al., 2025)'s official codebase and adhered to the official hyperparameter settings throughout, including: random batches of 15 questions for API proposal, GroundingDINO-SwinT-OGC (Liu et al., 2024) for object detection and UniDepth-v2-ViTS14 (Piccinelli et al., 2024) for depth estimation—the exact same tools used in our TVP implementation.

### B.2   TVP CONFIGURATIONS

We run the TVP pipeline on Omni3D-Bench (§3.1) with the following configurations:

For the main pipeline (Alg. 1), we process all $N = 501$ questions from the entire dataset over $T = 3$ iterations. During each iteration, we generate $m = 4$ program candidates per question and retrieve $k_{\max} = 3$ similar examples from the example library $\mathcal{E}$ using BGE-Large-En-v1.5 embeddings (Xiao et al., 2023) with an embedding similarity threshold $\tau_{\text{sim}} = 0.8$. Programs are accepted into the example library only if their quality score exceeds $\tau_q = 8.5$ on a 10-point scale. The tool abstraction

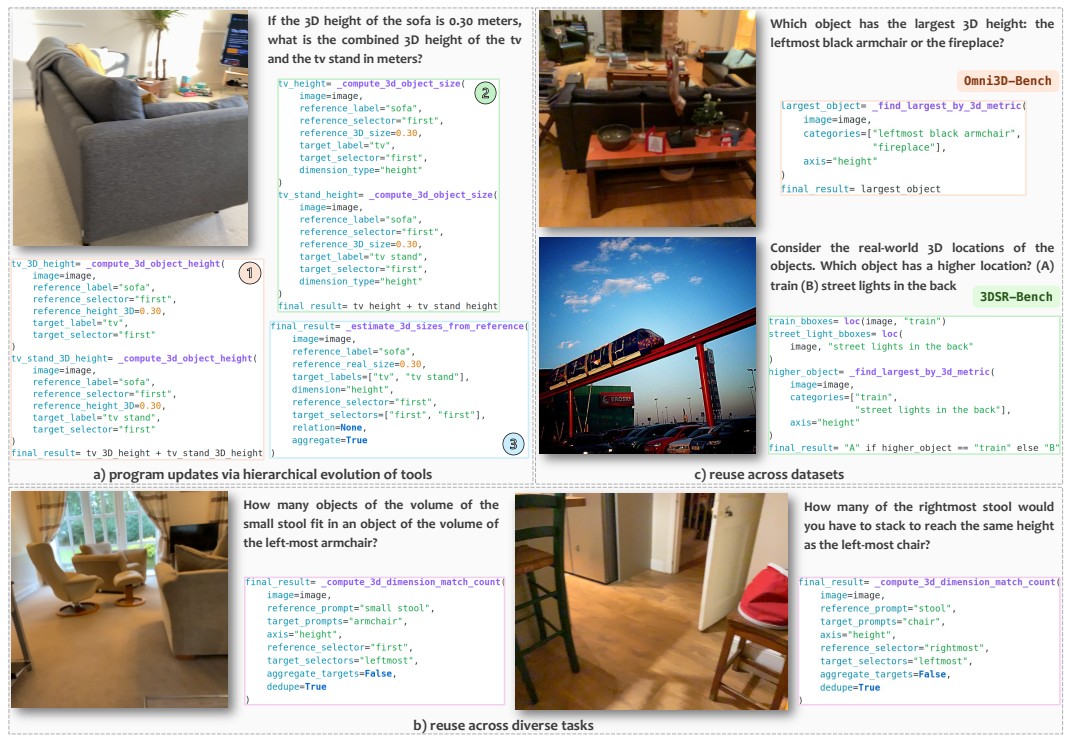

Figure 10: Transductively created tools enable hierarchical evolution and diverse reuse within and across benchmarks. (a) Programs simplify as tools evolve: the same spatial reasoning query is solved using progressively more general tools. (b) A single learned tool handles diverse dimension-matching problems within Omni3D-Bench. (c) Tools transfer to unseen benchmarks without modification, solving structurally similar problems in different contexts.

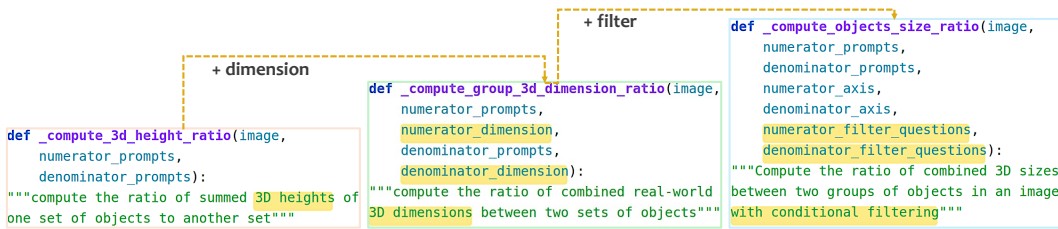

Figure 11: Hierarchical tool evolution through closed-loop refinement. TVP progressively abstracts more general tools through clustering (§2.3) and merging (§2.5)—each evolution subsuming the previous tool's functionalities.

process (Alg. 2) is triggered continuously after every $n_a = 1$ step (effectively at every step). We cluster examples using a similarity threshold of $\tau_{\text{sim}} = 0.8$ and require a minimum cluster size of $\tau_{\text{cluster}} = 4$ examples. Clusters with an abstraction potential score above $\tau_{\text{potential}} = 9.0$ are considered for tool creation. The validation process (Alg. 3) requires a minimum execution success rate of 100% and an overall correctness rate of at least $\tau_{\text{correct}} = 0.85$ (85%), where identical results are counted as correct and divergent results must be validated by the correctness validator ($\text{LLM}_{\text{correct}}$). Both abstraction and program rewriting allow up to $R_{\max} = 2$ and $R_{\text{rewrite}} = 2$ retry attempts respectively. Tool deduplication (Alg. 4) is also performed after every $n_d = 1$ step. Tools are considered duplicates when their similarity exceeds 0.95. The merge process allows up to $R_{\text{merge}} = 2$ retry attempts to create a unified tool that passes validation.

Throughout our experiments, we maintain a **uniform random seed** of 42 across all pipeline components, governing aspects such as datapoint order (discussed more in §C.1).

In our main experiments (Tab. 1), we employ GPT-4o as the backbone program generator ($LLM_{prog}$), and as the VLM-based quality judge ($LLM_{judge}$) & correctness validator ($LLM_{correct}$). We use the reasoning model o4-mini for clustering ($LLM_{cluster}$), abstraction ($LLM_{abstract}$), deduplication ($LLM_{dedup}$), merging ($LLM_{merge}$), and program rewriting tasks. In our ablations on the scaling behavior (Fig. 8), we switch to Qwen2.5-Coder-Instruct-7/14/32B (Hui et al., 2024) for the backbone program generation, and run TVP for $T = 1$ iteration. Results discussed in §3.1 demonstrate TVP's **robustness to the backbone LLM**, as well as the **clear scaling trend with model capacity**.

Unless required by specific reasoning models like o4-mini (temperature = $1.0$), LLM **temperatures** are set to $0.0$ for deterministic tasks (quality judgment and correctness validation), ensuring rigorous assessment; and $1.0$ for more creative tasks (program generation, abstraction, and rewriting), increasing the likelihood of finding better solutions.

In Fig. 5(a), we use McCabe's Cyclomatic Complexity Number (CCN) (McCabe, 1976) as the program complexity measure, computed via the Lizard library—following the practice in Yuan et al. (2023).

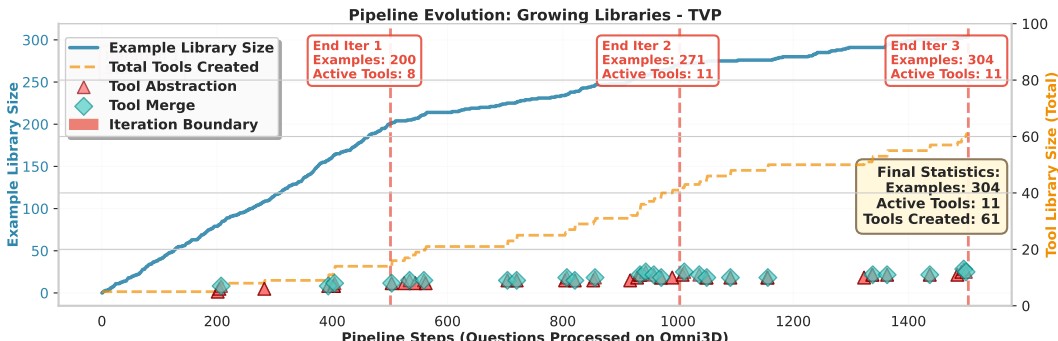

Figure 12: Evolution trajectory of TVP's dual libraries. The Example Library grows steadily to $304$ solutions while the Tool Library expands strategically—creating $61$ tools total but maintaining $11$ active abstractions through periodic merging.

### B.3 CRITERIA IN TVP'S JUDGE COMPONENTS

The program quality judge (§2.2) gates the admission to TVP's Example Library through evaluation across five comprehensive dimensions (as shown in Prompt 1): (1) 3D spatial understanding, following Marsili et al. (2025)'s official implementation for 3D concepts and definitions; (2) answer correctness with visual verification against the provided image; (3) appropriate program tool usage; (4) code quality including readability and efficiency; and (5) robustness to edge cases. These dimensions align with the critical requirements of both spatial reasoning and programming. The reliability of our quality judge is empirically validated in Tab. 1, where **enabling only the Example-Library in TVP already outperforms all baselines**. This demonstrates accurate admission of high-quality solutions in our Example Library that provide strong in-context examples.

The criteria for **tool abstraction potential** can be found in Prompt 2, which analyzes a group of program solutions clustered via question embeddings (embedding similarity is the first step of clustering, refer to §2.3). The abstraction potential focuses on general code abstraction requirements: (1) common computational patterns; (2) logical flow; (3) generalization capability; and (4) parameterization potential. We allow this flexibility in tool abstraction to **enable more diverse exploration of higher-level tools**, while still ensuring new tools' quality through the rigorous validation against all examples in the cluster before Tool Library admission (§2.3).

### B.4 COMPLEXITY RATING OF 3D SPATIAL REASONING QUESTIONS

In both our complexity-grouped evaluation illustrated in Figs. 6 and 7), and the curriculum-ordered TVP run discussed in §C.1, we use the **question complexity scores** rated via GPT-5 ("high" reasoning effort) with the prompt given in Prompt 3. The complexity rating evaluates questions along five axes, considering *e.g.*, 3D understanding; single-/multi-object relationships and multi-step reason-

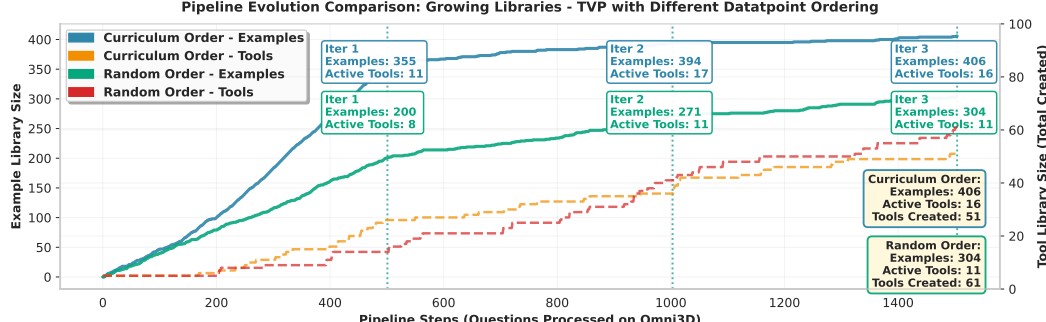

Figure 14: Pipeline evolution comparison between curriculum-ordered (as defined in §C.1) and random-ordered (as in our main experiments §3.1) datapoint processing. While curriculum ordering enables faster initial accumulation of examples and tools, random ordering ultimately creates more diverse tools through broader exploration of the problem space.

ing; cognitive and computational load. Based on these scores on the scale of 1.0–10.0, we partition questions into three complexity buckets: Easy (1–3), Medium (4–6), and Hard (7–10).

## C    ADDITIONAL EMPIRICAL ANALYSES AND DISCUSSION

### C.1    TVP'S RESILIENCE TO RANDOM DATAPOINT ORDERING

TVP is designed to operate on the fly without any dataset-specific priors, unlike previous methods such as Skillweaver (Zheng et al., 2025) that depends on human-defined curriculum for structured progression (see also §4.2). To validate our prior-free design choice, we compare the original TVP with **random ordering**, as used in our main experiments (§3.1) to **curriculum ordering** based on easy-to-hard progression through question complexity scores (details in §B.4).

Despite the intuition that starting experience with simpler problems, then gradually attempting harder problems seems a natural fit, we show in Fig. 13 that the overall performance is mostly on-par (both outperforming baselines), with the randomly-ordered TVP gradually getting better than the curriculum-ordered variant.

To understand this result, Fig. 14 reveals the evolution dynamics under both ordering strategies. The curriculum prior introduces two notable early-stage effects. First, it leads to **earlier example accumulation**: datapoints with similar complexity clustered together facilitate more frequent example retrieval, resulting in 355 accumulated examples versus 200 with random ordering at the end of iteration 1. Second, it promotes **earlier tool creation**, as similar and simpler examples form eligible clusters sooner, yielding 11 active tools compared to 8 with random ordering after the first iteration.

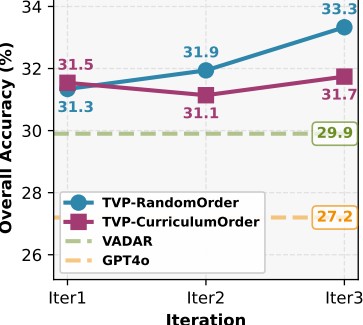

Figure 13: Performance comparison between TVP runs with random vs. curriculum-based datapoint ordering.

However, **random ordering proves more beneficial for sustained library growth**. By encountering datapoints of varying complexities and patterns throughout processing, TVP explores a more diverse solution space. Although initial accumulation may be slower, this diversity enables continued progression as both libraries capture richer patterns. By completion, random ordering generates 61 total tools compared to 51 with curriculum ordering, demonstrating the value of diverse exploration over structured progression.

The above analysis speaks for TVP's design choice of resilience to random datapoint ordering that enables **truly on-the-fly** operation without requiring any dataset-specific priors. The diverse exploration inherent to random ordering also fosters greater variety in accumulated experiences, leading

to **more comprehensive tool creation** that better covers the problem space. This mirrors human learning that benefits from exposure to varied challenges rather than strictly structured curricula.

## C.2 COMPUTATIONAL COST AND EFFICIENCY

We detail computational requirements for running TVP below:

**Cost structure and runtime.** TVP operates in two distinct stages with different cost profiles:

(1) *Building* TVP*'s dual libraries from scratch* involves processing the test set on the fly (Omni3D-Bench in §3.1), analogous to training a model. This stage requires approximately $80 per iteration with our GPT-4o + o4-mini configuration, equivalent to about $0.16 per question per iteration.

(2) *Applying* TVP*'s built libraries to solve questions* (SpatialScore-*Hard* collection in §3.2) has minimal cost, usually equivalent to a single GPT-4o call per query.

GPUs are strictly optional in both stages. When used, the system requires under 4GB VRAM only to store the basic vision tools (GroundingDINO (Liu et al., 2024) and UniDepth (Piccinelli et al., 2024)), which can also run on CPUs. The runtime for building TVP's dual libraries (analogous to training) stands at approximately 7 hours per iteration with our current implementation that executes programs sequentially.

**Efficiency optimizations.** We implement several strategies to improve TVP's cost-efficiency when building its dual libraries:

(1) *Early exit in tool validation*: Abstracted tools must achieve 100% execution success and 85% correctness on their validation cluster as per our current configurations (§B.2). For a cluster of *e.g.* 7 examples, validation exits early—thus avoiding unnecessary computation—when any one example fails execution (100% requirement not met); or when two fail the correctness check ($5/7 = 71\%$, drops below 85% pass rate)

(2) *Easy resumability*: We maintain comprehensive state checkpoints, supporting TVP's pause and resume at any point.

(3) *Embedding bank*: Since question embeddings remain unchanged, we keep a persistent storage of embedding vectors that enables simple lookup when retrieving (§2.2) or clustering examples (§2.3).

(4) *Parallel program generation and quality judge*: We generate program candidates in parallel and batch the quality judging for all valid candidates to reduce run-time.

## D PROMPT TEMPLATES

### Prompt 1: Quality Judge

```
You are an expert judge evaluating the quality of a program that solves a 3D spatial reasoning problem with
    tools (functions). Your task is to assess the program based on specific criteria and assign a quality
    rating from 1.0 to 10.0.

## TASK OVERVIEW
### Question
question

### Program to Evaluate
```python
program_code
```

### Execution Results
- **Status:** exec_status: success/failure
- **Final Answer:** answer_text
- **Tools Used:** used_tools
- **Execution Error:** execution_error if present

### Execution Namespace (All Variables)
execution_namespace_text

## EVALUATION FRAMEWORK
### Visual Evidence Verification
You are provided with an image of the scene. Use this visual evidence throughout your evaluation to:
- Verify the program's approach aligns with what's visible in the image
- Validate the answer's reasonableness based on visual proportions
```

```
- Check if 3D spatial relationships are correctly interpreted
- Confirm intermediate results match the visual scene
- Ensure all verification considers 3D content, not just 2D positioning

### Namespace Analysis Considerations
When reviewing the execution namespace, specifically check for:
- Correct object identification (right object selected from multiple candidates)
- Proper bounding box matching
- Expected intermediate calculations
- Appropriate object filtering
- Correct depth-based 3D conversions
- Mismatched bounding boxes or coordinates
- Unexpected intermediate calculation results
- Objects that should have been filtered but weren't

### 3D Spatial Concepts & Definitions
**Core Definitions:**
- **Coordinate system:** (width, height, length) = (x, y, z) axis
- **Depth:** Distance from camera (smaller depth = closer to camera)
- **2D measurements:** Size/distance in pixel space (image coordinates)
- **3D measurements:** Size/distance in real world
- **3D size formula:** `3D size = 2D size * depth`
- **2D distance formula:** Euclidean distance between object center coordinates: `((x1-x2)**2 + (y1-y2)**2)**
    0.5`
- **3D distance formula:** `3D_distance = (2D_distance**2 + (depth1 - depth2)**2)**0.5`
- **Distance to camera:** Simply the object's depth value

**Key Considerations:**
- 2D sizes are in pixel space. To convert to 3D size, multiply by depth
- Objects with same 2D dimensions but different depths have different 3D sizes
- 3D distance requires the Pythagorean formula combining 2D distance and depth difference – as defined above
- Center coordinates should determine "leftmost"/"rightmost"
- The `loc()` function should not handle compound descriptions – must locate base objects then filter for the
    desired condition
- All objects satisfying a condition must be checked, not just the first
- Multiple objects with same property values require proper tie-breaking
- Hypothetical object counts (e.g., "if a table has X legs") require counting actual objects in image

## RATING CRITERIA (1.0 – 10.0 Scale)
### 1. **3D Spatial Understanding**
- Properly converts between 2D and 3D measurements
- Correctly handles 3D size/distance calculation
- Correctly uses center coordinates for distance calculations and leftmost/rightmost determinations
- Interprets spatial relationships correctly (e.g., "largest" means 3D, not 2D)
- Answer is visually verifiable and reasonable

### 2. **Correctness and Visual Verification**
- Solves the problem correctly based on the actual image
- Aligns with visual evidence from the image
- Intermediate results are consistent with visible scene
- Spatial relationships match visual reality

### 3. **Tool Usage Efficiency**
- Uses appropriate tools for the task
- Leverages higher-level "learned" tools when suitable
- Avoids reimplementing existing functionality
- Note: Basic tools are acceptable when no higher-level tools fit

### 4. **Code Quality**
- Well-structured with clear variable names
- Follows tool usage patterns correctly
- Efficient without unnecessary operations
- Includes helpful comments

### 5. **Robustness and Edge Cases**
- Properly filters located objects for properties rather than using complex `loc()` queries
- Handles multiple objects with same property (proper tie-breaking)
- Manages empty lists and None values appropriately
- Manages container relationships (e.g., "in", "on") properly
- Includes appropriate error checking

## REQUIRED OUTPUT FORMAT
You MUST provide your response in exactly this format:

<rating>NUMBER</rating>
<reasoning>
[Detailed explanation covering:
- How visual evidence supports/contradicts the program's logic
- Specific strengths and weaknesses identified
- Analysis of 3D spatial reasoning approach
- Evaluation of intermediate execution results
- Missed opportunities to use available tools
- Overall assessment based on all criteria]
</reasoning>
Where NUMBER is a decimal between 1.0 and 10.0.

---

## APPENDIX: Available Tools Reference

The program had access to tool_counts tools total: num_basic_tools basic tools and num_created_tools learned
    tools.
```

```
### Basic Tools (Level 0)
tool_signature, tool_docstring

### Learned Tools (Level 1+)
tool_signature, tool_docstring
```

## Prompt 2: Abstraction Potential Analysis

```
You are an expert at analyzing visual reasoning programs to identify common patterns that could be abstracted
    into reusable functions.

## Your Task
Analyze num_cluster_examples visual reasoning examples to:
1. Identify common computational patterns across examples
2. Group them into clusters based on shared functionality
3. Rate each cluster's abstraction potential (0-10 scale)

## Examples to Analyze
examples (question, program solution)

## Clustering Criteria
Identify clusters based on:
1. **Common computational patterns** - e.g., finding largest/smallest, counting with conditions
2. **Similar operations sequence** - e.g., locate -> filter -> compute -> compare
3. **Shared logic structure** - e.g., iteration patterns, comparison logic
4. **Abstractable functionality** - can be parameterized into a reusable function

## Evaluation Requirements
### For Each Cluster Provide:
- **Example IDs** that belong to it
- **Common pattern** explanation
- **Parameters** that vary between examples
- **Abstraction potential rating** (0-10) based on:
  * How well the pattern generalizes
  * Parameter variability coverage
  * Clarity of the abstraction
  * Reusability across similar tasks
- **Reasoning** for the rating

### Critical Constraints
- **Each example must belong to exactly ONE cluster or be marked as unclustered**
- Focus on computational patterns, not surface similarities
- Only create clusters with strong shared patterns

## Response Format
Provide your analysis using this exact format. Include as many cluster blocks as needed, followed by an
    optional unclustered block:

```
<cluster>
<example_ids>[comma-separated list of example IDs]</example_ids>
<pattern>[Description of the common computational pattern]</pattern>
<parameters>[List of parameters that vary between examples]</parameters>
<abstraction_potential>[Rating from 0-10]</abstraction_potential>
<reasoning>[Explanation for the rating and how the pattern could be abstracted]</reasoning>
</cluster>

[Additional <cluster> blocks as needed...]

<unclustered>
<example_ids>[comma-separated list of example IDs that don't fit clusters]</example_ids>
<reasoning>[Explanation of why these examples don't cluster well]</reasoning>
</unclustered>
```

**Remember:** Every example ID must appear in exactly ONE cluster or in the unclustered group.
```

## Prompt 3: Question Complexity Rating

```
You are an expert in evaluating the complexity of 3D spatial reasoning questions. Your task is to assign a
    complexity score (1.0 - 10.0 scale) to a single question based on its inherent spatial reasoning
    difficulty.

## QUESTION TO EVALUATE

**Question:** question

**Answer Type:** answer type: float/integer/multiple-choice/etc.

## EVALUATION FRAMEWORK

### 1. **3D Spatial Reasoning Requirements**
- Does the question require understanding of 2D (pixel/image space) vs 3D (real-world) measurements?
- Does it involve depth understanding and distance-from-camera concepts?
- Does it require 3D size calculations or understanding that same 2D size at different depths means different
    3D sizes?
```

```
- Does it involve 3D distance calculations (combining 2D distance and depth differences)?
- Does it require converting between measurement spaces?

### 2. **Spatial Relationship Complexity**
- How many objects are involved in the spatial reasoning?
- Types of relationships:
  - Simple property identification (color, material, count)
  - Spatial relationships (distance, size comparison, relative position)
  - Complex spatial relationships (e.g., "to the right of X and behind Y")
- Does it require multi-step reasoning with intermediate conclusions?
- Comparative judgments vs. absolute measurements

### 3. **Cognitive Load and Constraints**
- Number of constraints or conditions that must be simultaneously satisfied
- Need to identify and distinguish between multiple candidate objects
- Hypothetical or conditional reasoning ("if X is Y meters, then...")
- Handling of multiple objects with potentially ambiguous descriptions
- Container relationships (objects "in" or "on" other objects)

### 4. **Calculation and Quantitative Complexity**
- Simple identification or counting vs. numerical calculations
- Ratio, proportion, or percentage calculations
- Multiple measurement comparisons
- Distance or size computations requiring formulas
- Precision requirements

### 5. **Answer Type Indicators**
- **yes/no (binary):** Often simpler verification tasks but can be complex depending on what's being verified
- **multiple choice (str with options):** Requires discrimination among bounded options
- **numerical (float/int):** Often requires precise calculations and measurements
- **open string:** May require identification and categorization

## COMPLEXITY SCORING GUIDELINES

Consider the full spectrum of complexity:

**Lower end:** Simple, direct questions requiring minimal spatial reasoning
- Single object property identification
- Basic counting
- Simple yes/no verification with clear criteria

**Middle range:** Moderate spatial reasoning and calculation
- Size or distance comparisons between pairs of objects
- Simple ratio calculations
- Object identification with multiple constraints
- Basic 2D-to-3D conversions

**Higher end:** Complex multi-step spatial reasoning
- Multiple object comparisons with numerous constraints
- Complex calculations involving combined measurements
- Hypothetical reasoning with conditional calculations
- Spatial relationships involving many objects with interdependencies
- Ratios of combined or derived quantities

Assign a score on the scale of 1.0 - 10.0 based on the question's position in this complexity spectrum.
     Consider ALL evaluation dimensions together.

## REQUIRED OUTPUT FORMAT

Provide your response in EXACTLY this format:

<score>X.X</score>
<reasoning>
[Detailed explanation covering:
- Which evaluation dimensions contribute most to complexity
- Specific aspects that increase or decrease difficulty
- Why this score is appropriate
- Key spatial reasoning challenges in the question]
</reasoning>

The score should be a decimal number between 1.0 and 10.0. Use your judgment to place the question
     appropriately on the complexity spectrum.
```

# E   COMPLETE TVP ALGORITHM

---

**Algorithm 1** Transductive Visual Programming (TVP) Pipeline

---

**Input:** Dataset $\mathcal{D} = \{(I_i, q_i)\}_{i=1}^N$ (images, questions)
1: **Initialize:** Example Library $\mathcal{E} \leftarrow \emptyset$, Tool Library $\mathcal{T} \leftarrow \{\text{predefined tools}\}$
2: **Parameters:** quality threshold $\tau_q$, abstraction interval $n_a$, deduplication interval $n_d$
3: **for** iteration $t = 1$ to $T$ **do**
4:     $n_q \leftarrow 0$                                                      $\triangleright$ processed question counter
5:     **for** each question $q_i \in \mathcal{D}$ **do**
6:         # Retrieve similar examples
7:         $\mathcal{E}_{\text{sim}} \leftarrow \text{RetrieveSimilar}(\mathcal{E}, q_i, k_{\max} = 3)$                   $\triangleright$ excluding $q_i$ itself
8:         # Generate program candidates
9:         $\mathcal{C} \leftarrow \emptyset$
10:         **for** $j = 1$ to $m$ **do**                               $\triangleright$ $m$ candidates per question
11:             $p_j \leftarrow \text{LLM}_{\text{prog}}(q_i, \mathcal{E}_{\text{sim}}, \mathcal{T})$           $\triangleright$ in-context learning with $\mathcal{E}_{\text{sim}}$
12:             $\mathcal{C} \leftarrow \mathcal{C} \cup \{p_j\}$
13:         **end for**
14:         # Execute and filter none results
15:         $\mathcal{C}_{\text{succ}} \leftarrow \emptyset$
16:         **for** each $p_j \in \mathcal{C}$ **do**
17:             $\text{result}_j, \text{namespace}_j \leftarrow \text{Execute}(p_j, I_i, \mathcal{T})$
18:             **if** $\text{result}_j \neq \text{None} \wedge \neg\text{error}$ **then**
19:                 $\mathcal{C}_{\text{succ}} \leftarrow \mathcal{C}_{\text{succ}} \cup \{(p_j, \text{result}_j, \text{namespace}_j)\}$
20:             **end if**
21:         **end for**
22:         **if** $\mathcal{C}_{\text{succ}} \neq \emptyset$ **then**
23:             # Judge quality
24:             **for** each $(p_j, \text{result}_j, \text{namespace}_j) \in \mathcal{C}_{\text{succ}}$ **do**
25:                 $\text{quality}_j \leftarrow \text{LLM}_{\text{judge}}(q_i, p_j, \text{namespace}_j, I_i)$     $\triangleright$ 1-10 scale, criteria in §B.3
26:             **end for**
27:             # Select best and update example library
28:             $p^*, \text{quality}^*, \text{namespace}^* \leftarrow \arg\max_j \text{quality}_j$
29:             **if** $\text{quality}^* \geq \tau_q$ **then**
30:                 $e \leftarrow \text{Example}(q_i, p^*, \text{quality}^*, \text{namespace}^*)$
31:                 **if** $\exists e' \in \mathcal{E}$ with $e'.q = q_i$ **then**          $\triangleright$ update existing entry in $\mathcal{E}$
32:                     **if** $\text{quality}^* > e'.\text{quality}$ **or** ($\text{quality}^* = e'.\text{quality}$ **and** different tools) **then**
33:                         $\mathcal{E} \leftarrow (\mathcal{E} \setminus \{e'\}) \cup \{e\}$
34:                     **end if**
35:                 **else**
36:                     $\mathcal{E} \leftarrow \mathcal{E} \cup \{e\}$                     $\triangleright$ add new entry to $\mathcal{E}$
37:                 **end if**
38:             **end if**
39:         **end if**
40:         $n_q \leftarrow n_q + 1$
41:         # Abstraction interval
42:         **if** $n_q \bmod n_a = 0$ **then**
43:             $\mathcal{T}, \mathcal{E} \leftarrow \text{AbstractTools}(\mathcal{E}, \mathcal{T})$                 $\triangleright$ §2.3, see Alg. 2
44:         **end if**
45:         # Deduplication interval
46:         **if** $n_q \bmod n_d = 0$ **then**
47:             $\mathcal{T}, \mathcal{E} \leftarrow \text{DeduplicateTools}(\mathcal{T}, \mathcal{E})$             $\triangleright$ §2.5, see Alg. 4
48:         **end if**
49:     **end for**
50: **end for**
51: **return** $\mathcal{E}, \mathcal{T}$

---

---

**Algorithm 2** AbstractTools - Tool Abstraction from Example Clusters

---

**Input:** Example Library $\mathcal{E}$, Tool Library $\mathcal{T}$
**Output:** $\mathcal{T}$ with new tools, $\mathcal{E}$ with updated records of abstracted entries
1: **Parameters:** similarity threshold $\tau_{\text{sim}} = 0.8$, cluster size threshold $\tau_{\text{cluster}} = 4$, abstraction potential threshold $\tau_{\text{potential}} = 9.0$
2: # Filter eligible examples
3: $\mathcal{E}_{\text{eligible}} \leftarrow \{e \in \mathcal{E} : e.\text{status} \neq \text{"abstracted"}\}$
4: # Initial cluster by similarity
5: $\mathcal{G} \leftarrow \text{ClusterBySimilarity}(\mathcal{E}_{\text{eligible}}, \tau_{\text{sim}})$
6: **for** each cluster $G \in \mathcal{G}$ with $|G| \geq \tau_{\text{cluster}}$ **do**
7:      # Analyze cluster for abstraction potential and common pattern
8:      pattern, potential $\leftarrow \text{LLM}_{\text{cluster}}(G)$
9:      **if** potential $\geq \tau_{\text{potential}}$ **then**                $\triangleright$ abstraction potential threshold, criteria in §B.3
10:          # Create tool with retry
11:          **for** retry = 1 to $R_{\text{max}}$ **do**
12:              $t \leftarrow \text{LLM}_{\text{abstract}}(G, \text{pattern}, \mathcal{T})$
13:              # Validate tool
14:              val $\leftarrow \text{ValidateTool}(t, G, \mathcal{T})$                $\triangleright$ see Alg. 3
15:              **if** val.passed **then**
16:                  $\mathcal{T} \leftarrow \mathcal{T} \cup \{t\}$
17:                  # Update examples with new tool
18:                  **for** each $e \in G$ with successful rewrite **do**
19:                      $e.\text{program} \leftarrow \text{val.rewritten}[e]$
20:                      $e.\text{status} \leftarrow \text{"abstracted"}$          $\triangleright$ mark as already abstracted
21:                      $e.\text{tools\_used} \leftarrow \{t\}$
22:                  **end for**
23:                  **break**
24:              **end if**
25:          **end for**
26:      **end if**
27: **end for**
28: **return** $\mathcal{T}, \mathcal{E}$

---

---

**Algorithm 3** ValidateTool - Two-Stage Tool Validation

---

**Input:** Tool $t$, Examples $G$, Tool Library $\mathcal{T}$
**Output:** Validation result with rewritten programs
 1: **Parameters:** correctness threshold $\tau_{\text{correct}} = 0.85$
 2: # Stage 1: Execution validation
 3: rewrites $\leftarrow \{\}$
 4: **for** each example $e \in G$ **do**
 5:     **for** retry $= 1$ to $R_{\text{rewrite}}$ **do**
 6:         $p' \leftarrow$ RewriteProgram($e$.program, $t$)
 7:         result$'$, namespace$' \leftarrow$ Execute($p'$, $e$.image, $\mathcal{T} \cup \{t\}$)
 8:         **if** result$' \neq$ None $\wedge \neg$error **then**
 9:             rewrites[$e$] $\leftarrow (p', \text{result}', \text{namespace}')$
10:             **break**
11:         **end if**
12:     **end for**
13:     **if** $e \notin$ rewrites **then**                  $\triangleright$ 100% execution success required, otherwise early exit
14:         **return** $\{\text{passed} : \text{False}, \text{errors} : \text{execution\_failures}\}$
15:     **end if**
16: **end for**
17: # Stage 2: Correctness validation for divergent results
18: correct $\leftarrow 0$, divergent $\leftarrow 0$
19: **for** each $e \in G$ with successful rewrite **do**
20:     **if** rewrites[$e$].result $\neq e$.result **then**
21:         divergent $\leftarrow$ divergent $+ 1$
22:         verdict $\leftarrow$ LLM$_{\text{correct}}(e, \text{rewrites}[e], e.\text{image})$
23:         **if** verdict $=$ "CORRECT" **then**
24:             correct $\leftarrow$ correct $+ 1$
25:         **end if**
26:     **end if**
27: **end for**
28: overall\_correct $\leftarrow (|G| - \text{divergent} + \text{correct})/|G|$
29: **if** overall\_correct $\geq \tau_{\text{correct}}$ **then**             $\triangleright$ 85% minimum correctness, §B.2
30:     **return** $\{\text{passed} : \text{True}, \text{rewrites} : \text{rewrites}\}$
31: **else**
32:     **return** $\{\text{passed} : \text{False}, \text{errors} : \text{correctness\_failures}\}$
33: **end if**

---

---

**Algorithm 4** DeduplicateTools - Merge Similar Tools

---

**Input:** Tool Library $\mathcal{T}$, Example Library $\mathcal{E}$
**Output:** $\mathcal{T}$ with merged tools, $\mathcal{E}$ with updated records of entries using affected tools
1: # Filter eligible tools
2: $\mathcal{T}_{\text{eligible}} \leftarrow \{t \in \mathcal{T} : t.\text{level} > 0 \wedge \neg t.\text{deprecated}\}$
3: # Find duplicate tools and merge strategies
4: $\mathcal{M} \leftarrow \text{LLM}_{\text{dedup}}(\mathcal{T}_{\text{eligible}})$                ▷ functional similarity $\geq 0.95$, §B.2
5: **for** each merge group $(M, \text{strategy}_M) \in \mathcal{M}$ **do**
6:      # Get examples using these tools
7:      $\mathcal{E}_M \leftarrow \{e \in \mathcal{E} : \exists t \in M, t \in e.\text{tools\_used}\}$
8:      **for** retry = 1 to $R_{\text{merge}}$ **do**
9:          $t_{\text{merged}} \leftarrow \text{LLM}_{\text{merge}}(M, \text{strategy}_M)$
10:          val $\leftarrow$ ValidateTool$(t_{\text{merged}}, \mathcal{E}_M, \mathcal{T})$          ▷ see Alg. 3
11:          **if** val.passed **then**
12:              $\mathcal{T} \leftarrow \mathcal{T} \cup \{t_{\text{merged}}\}$
13:              **for** each $t \in M$ **do**
14:                  $t.\text{deprecated} \leftarrow \text{True}$
15:                  $t.\text{reason} \leftarrow$ "Merged into $t_{\text{merged}}$"
16:              **end for**
17:              # Update examples
18:              **for** each $e \in \mathcal{E}_M$ with successful rewrite **do**
19:                  Update $e$ with merged tool
20:              **end for**
21:              **break**
22:          **end if**
23:      **end for**
24: **end for**
25: **return** $\mathcal{T}, \mathcal{E}$

---

