# OpenReview forum: "Transductive Visual Programming: Evolving Tool Libraries from Experience for Spatial Reasoning"
_ICLR.cc/2026/Conference — ICLR 2026 Poster_

### Official Review · Reviewer_44Xv · 2025-10-18

**Soundness:** 4
**Presentation:** 3
**Contribution:** 2
**Rating:** 6
**Confidence:** 4

**Summary:**

This paper introduces an advanced visual programming method for 3D spatial understanding. The method starts with basic tools and automatically generates, stores, and optimizes useful tools (functions) while observing and solving problems. The reported quantitative results on 3D spatial reasoning benchmarks surpass all baselines, demonstrating the powerful potential of the method.

**Strengths:**

- The paper is well-written. Figures 1 and 3 clearly present the motivation and method.

- The self-evolving pipeline is carefully designed. The composition of spatial functions is novel in this area.

- The quantitative results on the sampled SpatialScore-Hard are promising.

- The case study and evolution curve effectively demonstrate the impact of evolution

**Weaknesses:**

Visual programming for spatial reasoning has also been used in 3D vision tasks, such as 3D visual grounding [1, 2], and [2] also employs a self-evolving process for spatial reasoning. The authors may consider discussing these works in Section 4.1.

- In lines 159–161, the LLM explores $m$ programs, but only one final answer is provided. Is the final answer selected by the VLMs?

- In line 169, there is mention of a quality threshold for functions, but the metric for quality is not provided.

- In Figure 4(c), why is the performance with new APIs lower?

- There is no analysis of the additional token costs for evolution. Is the cost worthwhile for the observed improvement in overall performance (3.4% in absolute terms, approximately 10% in relative terms)?

- In line 165, does the "final answer" refer to the ground truth answer?

- There is no analysis of the accuracy of the VLM judge.

- Does the ordering of the Omni3D-Bench data used for codebase construction affect the codebase? Would altering the data order drastically impact the performance of the resulting code?

References:

[1] Visual Programming for Zero-shot Open-Vocabulary 3D Visual Grounding, in CVPR, 2024.

[2] Language-to-Space Programming for Training-Free 3D Visual Grounding, in EMNLP, 2025

**Questions:**

See the weaknesses section. If I have any misunderstandings, I would greatly appreciate the authors' clarification.

---

> ### Author Response · Authors · 2025-11-25
> **Response to Reviewer 44Xv (PART 1/3)**
>
> We thank the reviewer for the constructive feedback and address each concern below:
>
> **[Q1]:** How does the ordering of Omni3D-Bench data affect TVP system?
>
> **[A1]:** As we discuss in **§D.1**, our TVP -- despite processing a dataset on the fly -- is **highly resilient to random variations** of the datapoint order. This is a distinguishing factor from related work that relies on a strong curriculum prior (see §B).
>
> In §D.1, we run a variant where datapoints are ordered according to **an easy-to-hard curriculum** based on problem complexity scores on a scale of 1.0-10.0 (details in §C.4). Intuitively, starting with simpler problems, accumulating experiences, then gradually attempting harder problems seems a natural fit for our programming system when provided with datapoint complexity priors. As we show in **Fig.13**, the overall performance is, however, roughly on-par between curriculum-ordered and random-ordered (random seed=42 as in our main experiments) runs, **both outperforming baselines**; but the randomly-ordered TVP gradually gets better than the curriculum-ordered variant.
>
> To investigate this result further, **Fig.12** shows the evolution dynamics under both ordering strategies. We observe two notable early-stage effects introduced by the curriculum prior:
>
> 1. Earlier Example Accumulation: Datapoints ordered close together share similar complexity and are more likely similar to each other. This enables more frequent example retrieval as in-context guidance, leading to **earlier accumulation of examples** (355 vs. 200 at the end of iteration 1).
>
> 2. Earlier Tool Creation: Similar and easier examples accumulate first, forming earlier clusters eligible for tool abstraction. This produces **earlier growth of the Tool Library** (11 vs. 8 active tools at the end of iteration 1).
>
> However, we note that our original prior-free random ordering (as per our TVP design) is **more beneficial for sustained growth of both libraries**. As datapoints of various complexities and logics are randomly encountered by the TVP system, it enables more thorough exploration of diverse solutions to diverse question patterns. While experience accumulation and tool creation may be slower at the beginning, there is more room for progression as both libraries capture richer patterns. By the end of iterations, **more tools are created with random ordering** (61 vs. 51 total tools created). This effect mirrors human learning that benefits from exposure to varied random challenges rather than strictly structured curricula.
>
> In sum, we would argue for the randomness of datapoint order as per our TVP design for two benefits as evidenced above: (1) **truly on-the-fly operation** with no reliance on dataset priors, unlike curriculum-dependent previous work (§B), and (2) **more random exploration enables more diversity** in accumulated experience, leading to a richer tool library development.
>
> ---
>
> **[Q2]:** What are the evaluation criteria for Example Library admission?
>
> **[A2]:** We clarify the criteria in **§C.3**, and discuss them below:
>
> **Example Library Quality Criteria:** As shown in Prompt 1, program quality is judged based on:
>
> (1) 3D spatial understanding (3D concepts and definitions follow VADAR's official implementation),
>
> (2) answer correctness with visual verification,
>
> (3) program tool usage,
>
> (4) program code quality,
>
> (5) robustness to edge cases.
>
> These dimensions constitute a comprehensive suite of criteria for spatial reasoning, and cover **critical aspects of programming**. TVP's **solid performance even without tool abstraction** (see §3.1) demonstrates that the examples admitted to our library following this suite of evaluation are indeed of high quality, and thus provide effective in-context demonstrations.
>
> ---
>
> **[Q3]:** How accurate is the VLM judge?
>
> **[A3]:** The VLM quality judge is one automatic component in our TVP system design.
>
> The accuracy and reliability of our quality judge is indicated by the fact that even without tool abstraction -- relying only on the **Example Library alone** to provide in-context demonstrations (the Example-Lib-Only variant) -- our system **already outperforms all other baselines** (see Tab.1 and discussion in **§3.1**). This demonstrates that the examples selected by our VLM-judge are indeed of high quality, providing effective in-context demonstrations that lead to superior performance.
>
> We detail the evaluation criteria of our quality judge in §C.3. Our judge evaluates programs based on a comprehensive suite of dimensions, aligned with the 3D spatial reasoning requirements and critical aspects of programming, as also discussed in **[Q2/A2]**.
>
> Given the above evidence, we believe our VLM-judge to be trustworthy in serving its role in the TVP system.

---

> > ### Author Response · Authors · 2025-11-25
> > **Response to Reviewer 44Xv (PART 2/3)**
> >
> > **[Q4]:** What is the computational cost of running TVP?
> >
> > **[A4]:** We clarify runtime, cost, and memory requirements for the TVP pipeline in **§D.2**, and summarize details below:
> >
> > 1. Cost Breakdown: There are two distinct stages with TVP:
> >
> >     (1) Building dual libraries from scratch : Processing the test set on the fly (Omni3D-Bench in §3.1), **analogous to training a model**.
> >
> >     (2) Applying dual libraries: Using the built libraries to solve questions (SpatialScore-Hard collection in §3.2), **analogous to model inference**.
> >
> >     For applying TVP in inference, the cost per query is usually equivalent to a single GPT-4o program generation call.
> >
> >     For building TVP's libraries, we keep the total API cost at approximately \$80 per iteration with our GPT-4o + o4-mini configurations. GPUs are strictly optional. When used, the system requires under 4GB VRAM only to store the basic vision tools (GroundingDINO and UniDepth), which can also run on CPUs. The runtime for building TVP's dual libraries (analogous to training) stands at approximately 7 hours per iteration with our current implementation that executes programs sequentially. With parallel execution (discussed below), this can be further reduced.
> >
> > 2. **Efficiency Optimizations**: We implement several strategies to improve cost-efficiency of TVP during the library build-up stage:
> >
> >    - Early Exit in Validation: Abstracted tools must pass validation on a cluster of examples at 100% execution success and 85% validation-pass rate as per our current configurations (§C.2). For instance, on a cluster of 7 examples, validation exits early as soon as one example fails execution (100% requirement not met) or two examples fail the correctness validation (5/7 = 71%, drops below 85% pass rate), thus avoiding unnecessary computation.
> >
> >    - Easy Resumability: We maintain comprehensive checkpointing storage for the system run and support **resume at any point** when the system exits unexpectedly, making it convenient to pause and resume TVP whenever needed.
> >
> >    - Embedding Bank: Since question embeddings are unchanged, we maintain a storage of embedding vectors and replace the embedding process with **simple lookup** when retrieving or clustering examples.
> >
> >    - Parallel Program Generation and Quality Judge: We generate program candidates in parallel, and batch the quality judging for all valid candidates to reduce run-time.
> >
> >     Additional measures can be implemented for faster runtime, such as **parallel execution** of program candidates and parallel validation of all examples in tool validation. The latter may slightly increase API cost — compared to our current **early exit** strategy — but would further speed up the validation process.
> >
> > In sum, the cost for applying TVP in inference equals a single GPT-4o program generation call per query at minimal expense. Building TVP's dual libraries is also highly accessible, with cost at \\$80 per iteration, equivalent to \\$0.16 per question. We have also implemented multiple efficiency optimizations, with flexibility for even further improvements. We thus believe that cost and efficiency overall is **not a barrier to TVP's research and practical deployment**.

---

> ### Author Response · Authors · 2025-11-25
> **Response to Reviewer 44Xv (PART 3/3)**
>
> **[Q5]:** Several clarification questions.
>
> **[A5]:** (*Lines* refer to previous paper version in this Q&A)
>
> *1. In lines 159–161, the LLM explores multiple programs, but only one final answer is provided. Is the final answer selected by the VLM?*
>
> Yes, the program representing the final solution is the program candidate with successful execution and the highest score judged by the VLM-judge. If no program candidates execute successfully (in rare cases), we consider the final answer None, which is immediately considered wrong during evaluation.
>
> *2. In Figure 4(c), why is the performance with new APIs lower?*
>
> The datapoints using new APIs are not the same as those always using basic APIs. Problems that basic vision tools can solve represent a **simpler batch of questions** that are inherently easier to answer correctly.
>
> Fig.4(c) serves to show the progression of performance in both problem groups. We observe steady performance utilizing basic tools only (blue bars), while **increasingly better performance** for programs using the created tools (red bars), which indicates a progressively better tool repertoire created by TVP. The iterative improvement of our TVP system on especially hard questions is additionally discussed in **§3.1** (Fig.5 & Fig.6).
>
> *3. In line 165, does the "final answer" refer to the ground truth answer?*
>
> No, it refers to the answer produced by a program candidate. The quality judge evaluates the program solution based on its answer and execution trace, combined with visual verification. More details on the judge criteria are given in **§C.3**.
>
> We note that **no ground truth is ever accessed** during the TVP process, which is exactly the merit of our supervision-free self-evolving paradigm. Our system requires no labeling of data but works **on the fly**, improving itself through the agent's own exploration, experience accumulation, and tool evolution.
>
> ---
>
> **[Q6]:** Discussing more related work [Mi2025, Yuan2024]
>
> **[A6]:**
> Both papers [Mi2025, Yuan2024] contribute unique designs to programming with predefined toolsets for 3D visual tasks. We have added discussion of both in **§4.2**.
>
> **References:**
>
> [Mi2025] Language-to-Space Programming for Training-Free 3D Visual Grounding. EMNLP, 2025.
>
> [Yuan2024] Visual Programming for Zero-shot Open-Vocabulary 3D Visual Grounding. CVPR, 2024.

---

> > ### Comment · Reviewer_44Xv · 2025-11-26
> >
> > The authors have effectively addressed my concerns with sufficient experiments and explanations, demonstrating that TVP:
> >
> > - Does not rely on GT answer and only requires VLMs for verification. Additionally, the accuracy of VLM verification has been analyzed (Q2, Q3).
> >
> > - Is robust regarding the ordering of "training" data (referring to the construction and optimization of the tool library) (Q1).
> >
> > - The token costs are acceptable (Q4).
> >
> > - Demonstrates that the tool library's capabilities gradually improve as the input data increases (Q5).
> >
> > In summary, this paper is well-presented and complete, with comprehensive experiments. I believe it makes a significant contribution to the field of visual programming. I have raised my score accordingly.

---

> > > ### Author Response · Authors · 2025-11-27
> > >
> > > Thank you very much for reviewing our revision and raising your score! We are glad that our responses have addressed your concerns, and we deeply appreciate your constructive feedback.

---

### Official Review · Reviewer_fsyr · 2025-10-27

**Soundness:** 3
**Presentation:** 4
**Contribution:** 2
**Rating:** 6
**Confidence:** 4

**Summary:**

This paper introduces Transductive Visual Programming (TVP), a novel framework that enables visual reasoning systems to evolve reusable tool libraries from problem-solving experience. TVP adopts a transductive approach: it first solves problems using basic vision tools, then abstracts recurring solution patterns into higher-level functions grounded in actual use. The architecture maintains a dual-library design: an Example Library storing verified program solutions and a Tool Library storing learned abstractions. Through iterative cycles of example accumulation, clustering, abstraction, validation, and merging, TVP progressively refines its tools and produces more efficient, accurate programs. On Omni3D-Bench, TVP achieves clear performance gains, surpassing GPT-4o and VADAR. The evolved tools exhibit strong zero-shot generalization to unseen spatial reasoning benchmarks, demonstrating robust transferability across domains.

**Strengths:**

- **Conceptual originality:** The paper introduces transductive tool evolution, which learns abstractions from experience rather than induction before use. This represents a genuine conceptual advance in visual programming and aligns well with human-like skill acquisition.
- **Technical soundness:** The dual-library architecture and full algorithmic specification (program generation, clustering, abstraction, validation, and merging) are rigorous and clearly grounded. The validation mechanism ensures newly learned tools remain correct and reusable.
- **Strong empirical results:** TVP achieves clear and consistent gains over prior visual programming systems and even large VLMs, particularly on complex 3D spatial reasoning and zero-shot transfer tasks.

**Weaknesses:**

- **Limited scope of evaluation:** While visual programming was originally designed for 2D visual reasoning and perception tasks, this paper evaluates TVP only on 3D spatial reasoning. It remains unclear whether the proposed transductive abstraction also benefits conventional 2D visual reasoning benchmarks (e.g., MME, MMMU).
- **Heavy dependence on large proprietary models:** TVP’s components rely heavily on GPT-4o and its mini variants. It remains unclear how performance scales with smaller or open-source models, which may limit reproducibility and accessibility.
- **Computational overhead:** The pipeline includes iterative clustering, abstraction, validation, and merging — likely computationally expensive. The paper reports no quantitative analysis of time or resource cost, which would be important for assessing practicality.

**Questions:**

1. Does TVP’s method also improve performance on conventional 2D visual reasoning tasks?
2. How sensitive is TVP to the choice of backbone LLM? Would similar gains be observed when replacing GPT-4o with smaller or open-source models, and could the authors provide scaling trends or partial ablations in this direction?
3. Could the authors provide a quantitative estimate of TVP’s computational and memory cost per iteration, and clarify whether any optimizations were applied to make the system practically deployable?

---

> ### Author Response · Authors · 2025-11-25
> **Response to Reviewer fsyr (PART 1/2)**
>
> We thank the reviewer for the constructive feedback and address each concern below:
>
> **[Q1]:** How does the performance of TVP scale to the choice of backbone LLM?
>
> **[A1]:** We discuss TVP's robustness to backbone model choice with a scaling trend in **§3.1** (*"TVP is robust to backbone LLM choice, showing a clear scaling trend with model sizes"*) and **Fig.7**. Specifically, we evaluate open-source models of varying sizes for the backbone program generation, with representative Qwen2.5-Coder-Instruct model family.
>
> As shown in Fig.7, we observe a **clear scaling trend with model sizes**. Notably, using an open-source 32B model, TVP achieves performance close to GPT-4o-backed variant (30.7% vs. 31.3%), and surpasses the previous best baseline VADAR (29.9%) despite its more capable GPT-4o backbone. This result proves that our TVP **does not rely on proprietary-specific optimal LLMs** but can achieve strong performance with more accessible open-source alternatives.
>
> The fact that TVP performance scales clearly with model sizes, points to **significant future potential** of our transductive tool creation system, as foundation models with enhanced capabilities become available.
>
> In sum, TVP demonstrates strong robustness to backbone model choice, with an open-source model outperforming GPT-4o-backed previous baselines. The **clear scaling trend suggests TVP will continue to improve** with more capable LLMs, highlighting its future potential.
>
> ---
>
> **[Q2]:** Why not evaluate on conventional 2D visual reasoning benchmarks such as MME and MMMU?
>
> **[A2]:** We focus on 3D spatial reasoning because it requires programmatic logic and calculations beyond visual perception -- precisely where the family of visual programming systems contributes value (see the opening paragraph of **§1** & **§3.1**). As shown in **Fig.5**, even frontier VLMs struggle severally on the harder batch of 3D spatial reasoning questions.
>
> Traditional 2D visual reasoning benchmarks such as MME and MMMU largely involve visual understanding without requiring programmatic logic or symbolic calculations. For these benchmarks, direct VLM VQA already achieves strong performance (e.g. [current leaderboard of MME](https://huggingface.co/spaces/opencompass/open_vlm_leaderboard) shows accuracies approaching 90% overall). We believe there is less practical need for any programming system on these benchmarks, as they do not merit the value of programmatic reasoning over direct VQA.
>
> TVP is designed as a general framework. Given the right suite of initial tools, it evolves its experience (Example Library) and tool repertoire (Tool Library) for any domain. While we focus evaluation on 3D spatial reasoning in this paper, we will open-source our full system to **support community exploration** of TVP's adaptation to more diverse domains.

---

> > ### Author Response · Authors · 2025-11-25
> > **Response to Reviewer fsyr (PART 2/2)**
> >
> > **[Q3]:** What is the computational cost of running TVP?
> >
> > **[A3]:** We clarify runtime, cost, and memory requirements for the TVP pipeline in **§D.2**, and summarize details below:
> >
> > 1. Cost Breakdown: There are two distinct stages with TVP:
> >
> >     (1) Building dual libraries from scratch : Processing the test set on the fly (Omni3D-Bench in §3.1), **analogous to training a model**.
> >
> >     (2) Applying dual libraries: Using the built libraries to solve questions (SpatialScore-Hard collection in §3.2), **analogous to model inference**.
> >
> >     For applying TVP in inference, the cost per query is usually equivalent to a single GPT-4o program generation call.
> >
> >     For building TVP's libraries, we keep the total API cost at approximately \$80 per iteration with our GPT-4o + o4-mini configurations. GPUs are strictly optional. When used, the system requires under 4GB VRAM only to store the basic vision tools (GroundingDINO and UniDepth), which can also run on CPUs. The runtime for building TVP's dual libraries (analogous to training) stands at approximately 7 hours per iteration with our current implementation that executes programs sequentially. With parallel execution (discussed below), this can be further reduced.
> >
> > 2. **Efficiency Optimizations**: We implement several strategies to improve cost-efficiency of TVP during the library build-up stage:
> >
> >    - Early Exit in Validation: Abstracted tools must pass validation on a cluster of examples at 100% execution success and 85% validation-pass rate as per our current configurations (§C.2). For instance, on a cluster of 7 examples, validation exits early as soon as one example fails execution (100% requirement not met) or two examples fail the correctness validation (5/7 = 71%, drops below 85% pass rate), thus avoiding unnecessary computation.
> >
> >    - Easy Resumability: We maintain comprehensive checkpointing storage for the system run and support **resume at any point** when the system exits unexpectedly, making it convenient to pause and resume TVP whenever needed.
> >
> >    - Embedding Bank: Since question embeddings are unchanged, we maintain a storage of embedding vectors and replace the embedding process with **simple lookup** when retrieving or clustering examples.
> >
> >    - Parallel Program Generation and Quality Judge: We generate program candidates in parallel, and batch the quality judging for all valid candidates to reduce run-time.
> >
> >     Additional measures can be implemented for faster runtime, such as **parallel execution** of program candidates and parallel validation of all examples in tool validation. The latter may slightly increase API cost — compared to our current **early exit** strategy — but would further speed up the validation process.
> >
> > In sum, the cost for applying TVP in inference equals a single GPT-4o program generation call per query at minimal expense. Building TVP's dual libraries is also highly accessible, with cost at \\$80 per iteration, equivalent to \\$0.16 per question. We have also implemented multiple efficiency optimizations, with flexibility for even further improvements. We thus believe that cost and efficiency overall is **not a barrier to TVP's research and practical deployment**.

---

### Official Review · Reviewer_JN1p · 2025-10-30

**Soundness:** 3
**Presentation:** 3
**Contribution:** 2
**Rating:** 4
**Confidence:** 3

**Summary:**

This paper proposes Transductive Visual Programming (TVP), a framework for visual reasoning that dynamically creates and refines its own library of tools. The core idea is to learn from problem-solving experience. TVP maintains a "dual-library" system:

- An Example Library that stores (question, program, solution) tuples for high-quality solutions it has found.

- A Tool Library that contains callable functions (tools).

When faced with a new query, TVP retrieves similar examples from the Example Library to use as in-context demonstrations for generating a solution program. Critically, TVP periodically analyzes its Example Library, clusters similar solutions, and "transductively abstracts" recurring programming patterns into new, higher-level tools, which are then added to the Tool Library. This allows the system to evolve from basic tools to more complex, specialized, and reusable functions. The paper shows that this approach achieves SOTA on the Omni3D-Bench for 3D spatial reasoning and that the learned tools generalize well to unseen spatial benchmarks.

**Strengths:**

- The core idea of "transductive abstraction" from a library of successful solutions is elegant and well-motivated. It ensures that created tools are practically useful and grounded in experience, which is a clear advantage over VADAR's more speculative, question-based induction (as clearly shown in Fig 2).

- The zero-shot generalization results on the SpatialScore-Hard collection (Table 2, Fig 5) are a key strength. Showing that tools learned only on Omni3D-Bench are effective on completely different datasets (3DSR-Bench, SpatialSense, VG-Bench) is a powerful demonstration that the system is learning robust and reusable reasoning patterns.

- The paper includes a strong set of analyses, such as the reduction in program cyclomatic complexity (Fig 4a), the performance boost from using new tools (Fig 4c), and the visualization of the library evolution over time (Fig 6).

**Weaknesses:**

- The TVP framework itself is extremely complex and computationally expensive. For each query, it makes multiple LLM/VLM calls (retrieve, generate m programs, execute m programs, judge m programs). It then has a heavy, periodic maintenance loop that involves more LLM calls for clustering, abstraction, validation, and merging. This "meta-cost" of running the TVP framework is not discussed but seems prohibitively high, likely many times more expensive than just running a baseline model.

- As mentioned, the field of tool-use and tool-creation is moving very fast. This paper proposes a new way to make tools. While this is a good contribution, it's an improvement on an existing line of work (VisProg -> ViperGPT -> VADAR -> TVP). It's not clear that this is a fundamentally new direction for the field, especially when compared to orthogonal approaches like differentiable soft-logic (e.g., NePTune).

- The entire framework (generation, judgment, abstraction, merging) is orchestrated by powerful, closed-source models (GPT-4o and 4o-mini). This makes the system dependent on SOTA models and raises questions about its robustness. Would the framework collapse if these "meta-LLMs" were replaced with less capable open-source models? The quality of the "judge" and "abstractor" seems critical.

**Questions:**

- Could the authors please comment on the computational cost of the TVP framework? Specifically, how many total LLM/VLM calls (and of what type, e.g., GPT-4o vs 4o-mini) are required, on average, to process a single query (including the amortized cost of library maintenance)? This "meta-cost" seems like a major factor in its practical utility.

- The paper's related work cites other tool-creation work (e.g., Skillweaver, ASI) which also learn from "trajectories" or "experience." Could you elaborate on the key differences between TVP's "transductive abstraction" and the skill-discovery methods used in these agentic works?

- How robust is the TVP framework to the choice of its "meta-LLMs"? The system relies heavily on GPT-4o for crucial steps like quality judgment and program generation. If a weaker, open-source model (e.g., Llama3-8B) were used for these orchestration tasks, would the system still be able to successfully identify, abstract, and validate high-quality tools?

---

> ### Author Response · Authors · 2025-11-25
> **Response to Reviewer JN1p (PART 1/2)**
>
> We thank the reviewer for the constructive feedback and address each concern below:
>
> **[Q1]:** How does the performance of TVP scale to the choice of backbone LLM?
>
> **[A1]:** We discuss TVP's robustness to backbone model choice with a scaling trend in **§3.1** (*"TVP is robust to backbone LLM choice, showing a clear scaling trend with model sizes"*) and **Fig.7**. Specifically, we evaluate open-source models of varying sizes for the backbone program generation, with representative Qwen2.5-Coder-Instruct model family.
>
> As shown in Fig.7, we observe a **clear scaling trend with model sizes**. Notably, using an open-source 32B model, TVP achieves performance close to GPT-4o-backed variant (30.7% vs. 31.3%), and surpasses the previous best baseline VADAR (29.9%) despite its more capable GPT-4o backbone. This result proves that our TVP **does not rely on proprietary-specific optimal LLMs** but can achieve strong performance with more accessible open-source alternatives.
>
> The fact that TVP performance scales clearly with model sizes, points to **significant future potential** of our transductive tool creation system, as foundation models with enhanced capabilities become available.
>
> In sum, TVP demonstrates strong robustness to backbone model choice, with an open-source model outperforming GPT-4o-backed previous baselines. The **clear scaling trend suggests TVP will continue to improve** with more capable LLMs, highlighting its future potential.
>
> ---
>
> **[Q2]:** How does TVP stand out from related tool-use work?
>
> **[A2]:** We clarify additional details on the key differences between TVP and related work in **§B**, and summarize them below:
>
> **TVP vs. Skillweaver:**
>
> 1. Experience-Grounded vs. Inductive Tool Creation: Skillweaver is **purely inductive** (similar to VADAR): it speculatively proposes potentially useful functions before attempting any problem-solving experience, then synthesizes artificial test cases to validate these speculated functions. In contrast, TVP's tool creation is **grounded in actual problem-solving experience** by first accumulating experience of solving problems, then parameterizing this experience into new tools. This guarantees usefulness because each new tool encapsulates concrete program solution patterns.
>
> 2. Prior-Free vs. Curriculum-Reliance: Skillweaver follows a human-defined **curriculum** with a predefined task order (procedural → navigational → information-seeking) that fits the WebArena evaluation environment. This limits flexibility and adaptability. TVP is **prior-free** and allows **random ordering of datapoints**, making it more generally applicable. (See §D.1 for more discussion on TVP's resilience to randomness.)
>
> **TVP vs. ASI:**
>
> While ASI shares our insight of creating tools from the experience, TVP stands out in three aspects:
>
> 1. Clustering vs. Per-Episode Abstraction: ASI abstracts every episode individually, so **each new tool simply represents one single episode**. TVP abstracts a cluster of similar queries' solutions together, so that **each new tool generalizes over multiple example solutions**. This guarantees better reusability across a wider range of related problems.
>
> 2. Library Maintenance: ASI does not maintain the skill library or handle overlapping skills, which can lead to redundancy. TVP includes **explicit library maintenance** that merges similar skills (§2.4), keeping the tool library clean and concise.
>
> 3. Transductive Parameterization: ASI proposes multiple useful functions from one action trajectory rather than abstracting the whole solution itself. TVP directly parameterizes an entire cluster of program solutions into a single abstract tool.
>
> **TVP & NePTune:**
>
> **Orthogonal Contribution**: NePTune [Kamali2025] is an excellent contribution to the visual programming system family, with its specific focus on combining programmatic control flow with symbolic logic operators to enhance program expressiveness. However, TVP's focus lies in novel tool creation to enable more expressive programs through abstraction. These are orthogonal contributions.
>
> Additionally, NePTune is strictly contemporary work that came out on arXiv on Sep 30, 2025 -- after ICLR submission deadline.
>
> **References:**
>
> [Kamali2025] NePTune: A Neuro-Pythonic Framework for Tunable Compositional Reasoning on Vision-Language. Arxiv, 2025.

---

> > ### Author Response · Authors · 2025-11-25
> > **Response to Reviewer JN1p (PART 2/2)**
> >
> > **[Q3]:** What is the computational cost of running TVP?
> >
> > **[A3]:** We clarify runtime, cost, and memory requirements for the TVP pipeline in **§D.2**, and summarize details below:
> >
> > 1. Cost Breakdown: There are two distinct stages with TVP:
> >
> >     (1) Building dual libraries from scratch : Processing the test set on the fly (Omni3D-Bench in §3.1), **analogous to training a model**.
> >
> >     (2) Applying dual libraries: Using the built libraries to solve questions (SpatialScore-Hard collection in §3.2), **analogous to model inference**.
> >
> >     For applying TVP in inference, the cost per query is usually equivalent to a single GPT-4o program generation call.
> >
> >     For building TVP's libraries, we keep the total API cost at approximately \$80 per iteration with our GPT-4o + o4-mini configurations. GPUs are strictly optional. When used, the system requires under 4GB VRAM only to store the basic vision tools (GroundingDINO and UniDepth), which can also run on CPUs. The runtime for building TVP's dual libraries (analogous to training) stands at approximately 7 hours per iteration with our current implementation that executes programs sequentially. With parallel execution (discussed below), this can be further reduced.
> >
> > 2. **Efficiency Optimizations**: We implement several strategies to improve cost-efficiency of TVP during the library build-up stage:
> >
> >    - Early Exit in Validation: Abstracted tools must pass validation on a cluster of examples at 100% execution success and 85% validation-pass rate as per our current configurations (§C.2). For instance, on a cluster of 7 examples, validation exits early as soon as one example fails execution (100% requirement not met) or two examples fail the correctness validation (5/7 = 71%, drops below 85% pass rate), thus avoiding unnecessary computation.
> >
> >    - Easy Resumability: We maintain comprehensive checkpointing storage for the system run and support **resume at any point** when the system exits unexpectedly, making it convenient to pause and resume TVP whenever needed.
> >
> >    - Embedding Bank: Since question embeddings are unchanged, we maintain a storage of embedding vectors and replace the embedding process with **simple lookup** when retrieving or clustering examples.
> >
> >    - Parallel Program Generation and Quality Judge: We generate program candidates in parallel, and batch the quality judging for all valid candidates to reduce run-time.
> >
> >     Additional measures can be implemented for faster runtime, such as **parallel execution** of program candidates and parallel validation of all examples in tool validation. The latter may slightly increase API cost — compared to our current **early exit** strategy — but would further speed up the validation process.
> >
> > In sum, the cost for applying TVP in inference equals a single GPT-4o program generation call per query at minimal expense. Building TVP's dual libraries is also highly accessible, with cost at \\$80 per iteration, equivalent to \\$0.16 per question. We have also implemented multiple efficiency optimizations, with flexibility for even further improvements. We thus believe that cost and efficiency overall is **not a barrier to TVP's research and practical deployment**.

---

### Official Review · Reviewer_uXe8 · 2025-11-02

**Soundness:** 2
**Presentation:** 2
**Contribution:** 2
**Rating:** 4
**Confidence:** 3

**Summary:**

This paper presents Transductive Visual Programming (TVP), an innovative framework that enables visual language models to evolve by learning reusable tools from their own problem-solving experience.

The method’s dual-library closed-loop design is conceptually clear, technically comprehensive, and contributes a structured approach to self-improving reasoning in LLM-based systems.

While the framework is well-presented and methodologically sound, key evaluation details (e.g., scoring criteria, abstraction prompts, complexity metrics, and computational cost) remain under-specified.

Empirically, the performance gains over ICL baselines are modest, and the claimed generalization largely reflects in-domain transfer rather than genuine out-of-distribution generalization.

Overall, TVP is a promising and well-presented system paper: its transductive framework and dual-library design are conceptually valuable and clearly executed. Despite under-specified evaluation details and modest gains, I lean weak accept, contingent on clarifications.

**Strengths:**

Innovative and Well-Structured Framework

+ The paper introduces Transductive Visual Programming (TVP), a novel and conceptually original framework that enables a model to iteratively learn reusable tools from its own problem-solving experience. Its dual-library closed-loop design (Example–Tool Library) is systematic and complete, effectively realizing a self-improving learning cycle.

Strong presentation quality
+ The paper is clearly written, logically organized, and well-illustrated with informative figures and detailed algorithms, making the methodology easy to follow.

**Weaknesses:**

Lack of Transparency in Evaluation Mechanisms

The evaluation procedures governing both the Example Library and the Tool Library are under-specified, which raises concerns about reproducibility and interpretability. Specifically:
+ Unclear criteria for Example Library admission.
Although the paper states that a VLM judge scores each generated program and admits examples whose quality exceeds a threshold of τq = 8.5, it does not define the concrete scoring dimensions—such as logical correctness, semantic relevance, visual consistency, or execution success. The basis for selecting τq (e.g., validation tuning versus heuristic choice) is also not explained. Moreover, no analysis is provided regarding how scores vary across task types or problem complexity, leaving the quality control process for examples largely opaque.
+ Opaque evaluation of tool abstraction potential.
The paper introduces an LLM-based cluster analyzer that outputs a textual “pattern” and a numeric “potential” score, using τpotential = 9.0 as a threshold for initiating tool abstraction. However, the work omits any description of how this score is computed, the intended scale, or the prompts used to elicit it. The rationale behind the chosen threshold is also absent. If this potential measure relies solely on LLM-as-judge scoring, it is likely susceptible to style bias and semantic drift, undermining objectivity.

Limited Empirical Gains and Undefined Complexity Metric

While the paper claims that TVP achieves improved performance and reduced program complexity through iterative transductive learning, the empirical evidence supporting these claims appears limited and partially confounded.
+ Marginal performance improvement over ICL baselines.
The main results show that the Example Library-only configuration (essentially an ICL baseline) already achieves 31.7% overall accuracy, while the full TVP framework after three iterations reaches 33.3%. This modest +1.6% gain raises doubts about whether the improvement truly stems from the proposed abstraction mechanism, or instead from the growing in-context example set providing stronger template guidance to the LLM. The observed benefits may therefore largely reflect in-context pattern imitation rather than genuine tool learning. Moreover, performance surpasses the ICL baseline only at iteration 3, when computational cost and LLM usage are substantially higher—yet the paper offers no analysis of cost-effectiveness or scaling trade-offs.
+ Undefined program complexity metric.
The paper reports that “program cyclomatic complexity decreases from 3.0 to 1.0,” using this as evidence that TVP learns simpler, higher-level abstractions. However, no formal definition or computation method for this complexity measure is provided. It is unclear whether this refers to classical McCabe complexity, the number of function calls, or another proxy metric. Without such clarification, the claimed reduction in complexity cannot be meaningfully interpreted or independently verified.

High Computational Cost and Unanalyzed Efficiency

+ TVP surpasses the ICL-only baseline only at iteration 3, implying that multiple costly iterations are required for modest gains (+1.6 pp). Yet the paper provides no analysis of runtime, token usage, or cost. Given that each iteration repeatedly invokes GPT-4o for program generation, judging, abstraction, and validation, the overall expense is likely high. Without quantitative efficiency reporting, it is unclear whether the observed improvement justifies the computational overhead or scales beyond small benchmarks.

**Questions:**

Evaluation standards for Example and Tool Libraries

+ Could the authors clarify the evaluation criteria for Example Library admission and Tool Library abstraction? Specifically, what dimensions does the VLM judge consider when scoring examples (e.g., logical correctness, semantic relevance, visual consistency)?
+ How was the threshold τq = 8.5 chosen—through tuning, validation, or heuristic selection? Similarly, for the Tool Library, how is abstraction potential measured, what prompts are used, and on what basis was τpotential = 9.0 determined?
+ If both evaluations rely solely on LLM-as-judge scoring, how do the authors control for potential bias or inconsistency across runs?

Empirical significance and complexity metric

+ The improvement over the ICL baseline is relatively small (+1.6 pp) and only appears after three iterations. Could the authors provide more evidence that the observed gains stem from true tool abstraction rather than ICL-style template learning?
+ Also, please clarify how program complexity is computed (e.g., McCabe complexity, function calls, or another proxy metric), and explain why its reduction should indicate better abstraction quality.

Computational efficiency

+ Since performance exceeds the ICL baseline only at iteration 3, what is the computational cost of running multiple iterations?
+ Please report runtime, token usage, or cost per iteration, and discuss whether the modest gain justifies the overall expense or scales to larger datasets.

---

> ### Author Response · Authors · 2025-11-25
> **Response to Reviewer uXe8 (PART 1/3)**
>
> We thank the reviewer for the constructive feedback and address each concern below:
>
> **[Q1]:** Compare and explain the improvement of TVP full system over the Example-Library-only variant.
>
> **[A1]:** We provide an in-depth analysis in **§3.1** (*"Tool abstraction facilitates progressive self-improvement, especially on hard problems"*), supported by additional evidence shown in **Fig.5** & **Fig.6**. Below is a summary of our findings:
>
> 1. **The Example Library is an essential component of our TVP design**, accumulating the agent's own problem-solving experience that forms the foundation for developing tools and guiding future tool use. Exactly this experience-grounded approach distinguishes our transductive tool creation from inductive methods in prior works. Importantly, we provide no "gold" examplars to guide program generation with strong teacher priors. All examples stem from **the agent's own experience**, with in-context learning (ICL) serving simply as the implementation choice for utilizing this **experiential memory**. The fact that using the agent's own explored experience (i.e. the Example-Lib-Only variant) already outperforms previous baselines **demonstrates the quality of our example library**, enabled by our example library admission design (details in §C.3).
>
> 2. Tool Abstraction Enables Progressive Self-Improvement: Unlike the Example-Lib-Only variant which maintains largely unchanged performance across iterations (31.7%→31.5%→31.5%), the full TVP system with active Tool Library development leads to **clear progressive improvement** (31.3%→31.9%→33.3%). As also illustrated in Fig.4(c), datapoints using basic tools (i.e. no created tools) show essentially unchanged performance across iterations.
>
>     The progressive improvement stems from the **closed-loop design** of our system (see Fig.1): abstracted tools encapsulate past experience and enable better future programs, which become better examples, from which better tools can be abstracted. Fig.4(b) demonstrates this effect, showing +3.4% improvement when programs transition from basic tools to higher-level tools. Without tool creation in the loop, this self-improving cycle is weakened.
>
> 3. Tool Abstraction Shows Strongest Benefits on Hard Questions: To demonstrate where tool abstraction provides the most value, we analyze performance across question complexity levels. We use the most frontier GPT-5 (reasoning effort=high) to rate spatial reasoning questions on a 1.0-10.0 complexity scale (details in §C.4), then divide questions into three groups: Easy (1-3), Medium (4-6), Hard (7-10).
>
>     **Fig.5** shows accuracy across methods for different complexity levels. TVP (Full) **stands out as best-performing** on both Easy and Hard batches. For easy questions, thoroughly validated created tools avoid potential reimplementation errors, leading to more stable performance. For harder questions, created tools provide **simpler solution steps** that eliminate complicated logic, and ease the reasoning burden, as illustrated in the example in Fig.10(b), where a new tool serves as a convenient step in the program for a more complicated problem.
>
>     **Fig.6** shows the performance delta between TVP (Full) and TVP (Example-Lib-Only) across iterations for each complexity level. **On the hardest batch, TVP (Full) shows the most significant improvement**, starting at -4.5% relative to Example-Lib-Only in iteration 1, but ultimately surpassing it by +6.7% in iteration 3. This demonstrates the effect of tool abstraction parameterizing past experience into simple function calls, which **reduces the reasoning burden** for especially hard questions.
>
> In sum:
>
> - The Example Library provides an essential foundation for accumulating experience, which is core to our transductive method. When implemented via ICL, this experience already outperforms baselines, evidencing the quality of TVP's experience.
>
> - When tool creation is enabled, it encapsulates experience into simple function calls, bringing benefits most evident in: (1) **progressive improvement across iterations**, and (2) **substantial gains on the hardest batch of problems**.
>
> These aspects combined speak for the effectiveness of our transductive tool creation with the **closed-loop self-improving** capacity, which genuinely eases problem solving rather than mere in-context template learning.

---

> > ### Author Response · Authors · 2025-11-25
> > **Response to Reviewer uXe8 (PART 2/3)**
> >
> > **[Q2]:** What are the evaluation criteria for Example Library admission and Tool Library abstraction, and how were the thresholds determined?
> >
> > **[A2]:** We clarify the criteria in **§C.3**, and discuss them along with the threshold choices below:
> >
> > **Example Library Quality Criteria:** As shown in Prompt 1, program quality is judged based on:
> >
> > (1) 3D spatial understanding (3D concepts and definitions follow VADAR's official implementation),
> >
> > (2) answer correctness with visual verification,
> >
> > (3) program tool usage,
> >
> > (4) program code quality,
> >
> > (5) robustness to edge cases.
> >
> > These dimensions constitute a comprehensive suite of criteria for spatial reasoning, and cover **critical aspects of programming**. TVP's **solid performance even without tool abstraction** (see §3.1) demonstrates that the examples admitted to our library following this suite of evaluation are indeed of high quality, and thus provide effective in-context demonstrations.
> >
> > **Tool Abstraction Potential Criteria:** The tool abstraction criteria can be found in Prompt 2, which analyzes a group of program solutions clustered via question embeddings (embedding similarity is the first step of clustering, see §2.3). The abstraction potential focuses on general code abstraction requirements:
> >
> > (1) common computational patterns,
> >
> > (2) logical flow,
> >
> > (3) generalization capability,
> >
> > (4) parameterization potential.
> >
> > We allow this flexibility in tool abstraction to **enable more diverse exploration of higher-level tools**, while still ensuring new tools' quality through the rigorous validation against all examples in the cluster before Tool Library admission (refer to §2.3).
> >
> > **Threshold Selection:** The example quality threshold = 8.5 and abstraction potential threshold = 9.0 were empirically set -- quite analogous to how hyperparameters are configured in model training.
> > The example quality threshold = 8.5 controls the size and overall reliability of the Example Library, which is the foundation for our experience-based tool creation. As shown in Fig.9 (evolution log of TVP's growing libraries), **200 out of 501 program solutions (40%)** are included in the Example Library in the first iteration. This relatively selective standard ensures high-quality while maintaining sufficient number of examples for meaningful abstraction.
> > The abstraction potential threshold = 9.0 similarly facilitates tool creation that **truly implements a common function** parameterizing the cluster of programs.
> >
> > In our effort for reproducibility, we will open-source an easy-to-configure codebase of TVP, and hope this enables community exploration of hyperparameter configurations best suited to their specific needs.
> >
> > ---
> >
> > **[Q3]:** How is program cyclomatic complexity computed, and why does its reduction indicate better program quality?
> >
> > **[A3]:** We clarify the cyclomatic complexity metric and its interpretation below, and add these details to **§3.1** & **§C.2** of the revised paper.
> >
> > We use McCabe's Cyclomatic Complexity Number (CCN) [McCabe1976], computed via the [Lizard](https://github.com/terryyin/lizard) python library ([documents](https://github.com/terryyin/lizard/blob/master/theory.rst)), following the practice in related work CRAFT (ICLR, 2024) [Yuan2024].
> >
> > Program complexity is a widely-used metric for evaluating **program compression** in library learning and code generation research [Yuan2024, Grand2024, Bowers2023]. The argument is shared that the reduction of program complexity is a key benefit of effective library learning.
> >
> > In TVP, the reduction from average CCN=3.5 to average CCN=2.88 (-17.7%) and median CCN=3.0 to median CCN=1.0 indicates that our abstracted tools enable **simple function calls** that encapsulate otherwise more complicated code logic. The benefits of complexity reduction also manifest in two aspects:
> >
> > 1. Reduced Reimplementation Errors: Less repetitive reimplementation of the same branching logic enables higher efficiency and **reduced likelihood of potential errors** in reimplementing the code.
> >
> > 2. Improved Readability: The complexity reduction leads to **better interpretability of generated programs**, as a single function call replaces otherwise whole paragraphs of code (see qualitative examples in Fig.10).
> >
> > **References:**
> >
> > [McCabe1976] A complexity measure. IEEE Transactions on Software Engineering, 1976.
> >
> > [Yuan2024] CRAFT: Customizing LLMs by Creating and Retrieving from Specialized Toolsets. ICLR, 2024.
> >
> > [Grand2024] LILO: Learning Interpretable Libraries by Compressing and Documenting Code. ICLR, 2024.
> >
> > [Bowers2023] Top-down synthesis for library learning. POPL, 2023.

---

> > > ### Author Response · Authors · 2025-11-25
> > > **Response to Reviewer uXe8 (PART 3/3)**
> > >
> > > **[Q4]:** How do we control for potential inconsistency in LLM calls across runs?
> > >
> > > **[A4]:** We provide detailed configurations including LLM call hyperparameters in **§C.2**, and clarify our reproducibility measures below:
> > >
> > > 1. Temperature Settings: We strategically set temperature values based on each component's purpose. For program generation with GPT-4o, we use temperature=1.0 to enable **diverse exploration** of program candidates, thus increasing the likelihood of finding better solutions. For program quality judging and tool abstraction validation, we use temperature=0.0 to ensure **rigorous evaluation** with minimal variation across runs. Additionally, we use a **uniform random seed=42** throughout the experiments.
> > >
> > > 2. TVP's Inherent Resilience to Random Variations: TVP's design **inherently minimizes random inconsistency**, which is a key distinction from related works (see detailed discucsion in §B & §D.1). Because TVP creates tools from clusters of similar examples rather than single episodes, **individual variations in LLM calls are averaged out** across the cluster. Even if individual examples may vary, the abstracted tool still captures the common pattern across the cluster. Our active dual-library maintenance (§2.4) further ensures that occasionally varying tool generations do not compromise the overall library construct.
> > >
> > > ---
> > >
> > > **[Q5]:** What is the computational cost of running TVP?
> > >
> > > **[A5]:** We clarify runtime, cost, and memory requirements for the TVP pipeline in **§D.2**, and summarize details below:
> > >
> > > 1. Cost Breakdown: There are two distinct stages with TVP:
> > >
> > >     (1) Building dual libraries from scratch : Processing the test set on the fly (Omni3D-Bench in §3.1), **analogous to training a model**.
> > >
> > >     (2) Applying dual libraries: Using the built libraries to solve questions (SpatialScore-Hard collection in §3.2), **analogous to model inference**.
> > >
> > >     For applying TVP in inference, the cost per query is usually equivalent to a single GPT-4o program generation call.
> > >
> > >     For building TVP's libraries, we keep the total API cost at approximately \$80 per iteration with our GPT-4o + o4-mini configurations. GPUs are strictly optional. When used, the system requires under 4GB VRAM only to store the basic vision tools (GroundingDINO and UniDepth), which can also run on CPUs. The runtime for building TVP's dual libraries (analogous to training) stands at approximately 7 hours per iteration with our current implementation that executes programs sequentially. With parallel execution (discussed below), this can be further reduced.
> > >
> > > 2. **Efficiency Optimizations**: We implement several strategies to improve cost-efficiency of TVP during the library build-up stage:
> > >
> > >    - Early Exit in Validation: Abstracted tools must pass validation on a cluster of examples at 100% execution success and 85% validation-pass rate as per our current configurations (§C.2). For instance, on a cluster of 7 examples, validation exits early as soon as one example fails execution (100% requirement not met) or two examples fail the correctness validation (5/7 = 71%, drops below 85% pass rate), thus avoiding unnecessary computation.
> > >
> > >    - Easy Resumability: We maintain comprehensive checkpointing storage for the system run and support **resume at any point** when the system exits unexpectedly, making it convenient to pause and resume TVP whenever needed.
> > >
> > >    - Embedding Bank: Since question embeddings are unchanged, we maintain a storage of embedding vectors and replace the embedding process with **simple lookup** when retrieving or clustering examples.
> > >
> > >    - Parallel Program Generation and Quality Judge: We generate program candidates in parallel, and batch the quality judging for all valid candidates to reduce run-time.
> > >
> > >     Additional measures can be implemented for faster runtime, such as **parallel execution** of program candidates and parallel validation of all examples in tool validation. The latter may slightly increase API cost — compared to our current **early exit** strategy — but would further speed up the validation process.
> > >
> > > In sum, the cost for applying TVP in inference equals a single GPT-4o program generation call per query at minimal expense. Building TVP's dual libraries is also highly accessible, with cost at \\$80 per iteration, equivalent to \\$0.16 per question. We have also implemented multiple efficiency optimizations, with flexibility for even further improvements. We thus believe that cost and efficiency overall is **not a barrier to TVP's research and practical deployment**.

---

### Author Response · Authors · 2025-12-03
**Summary of Reviews and Rebuttal**

We thank all reviewers and the Area Chairs for reviewing our paper.

Here, we summarize the review questions and how we addressed them in the revised paper, and highlight the strengths of our work unanimously acknowledged by reviewers.

---

1. **Strengths of tool abstraction compared to example-library-only variant** (uXe8 [Q1]):

   Refer to Line 299-346 (§3.1), with key evidence in Fig.5 & Fig.6.
   Active tool creation enables progressive self-improvement across iterations, with strongest gains on the hardest questions.


2. **Performance scaling with the choice of backbone LLM** (JN1p [Q1], fsyr [Q1]):

   Refer to Line 367-377 (§3.1), with key evidence in Fig.7.
   TVP scales clearly and consistently with model capacity, where a 32B open-source model backbone reaches competitive performance, already surpassing prior baselines.


3. **Robustness to random variations in data processing order** (44Xv [Q1]):

   Refer to Line 804-856 (§D.1), with key evidence in Fig.12 & Fig.13.
   TVP is highly resilient to random variations—even a predefined easy-to-hard curriculum performs roughly on-par with random ordering.
   We further investigate why random ordering gradually outperforms the curriculum (Fig.13): the prior-free order enables more diverse exploration, supporting sustained library growth.


4. **Computational cost analysis**  (uXe8 [Q5], JN1p [Q3], fsyr [Q3], 44Xv [Q4]):

   Refer to Line 859-889 (§D.2).
   TVP is highly accessible in terms of cost, and we have implemented optimizations for efficiency.
   Cost is not a barrier to TVP's research or practical deployment.


5. **Discussing more related work and how TVP stands out** (JN1p [Q2], 44Xv [Q6]):

   Refer to revision in §4.2 and the additional §B, where we discuss in depth how TVP introduces a fundamentally novel paradigm different from representative prior work.


6. Clarifications:

   - **Evaluation criteria in TVP's judge components**: Line 773-789 (§C.3)

   - **Program compression** measured via McCabe's cyclomatic complexity: Line 267-298 (§3.1), Line 767-769 (§C.2)

---

In addition to addressing the reviewer questions, we highlight the key strengths of our paper unanimously praised by reviewers:


1. **Novel and well-motivated framework** around the core idea of "transductive abstraction", representing **a genuine conceptual advance** in visual programming (uXe8, JN1p, fsyr, 44Xv).


2. **Strong empirical results and zero-shot generalization** performance validating that **TVP learns truly robust and reusable patterns** (JN1p, fsyr, 44Xv).


3. **Comprehensive empirical analyses** demonstrating various facets of TVP's working mechanisms (JN1p, 44Xv)


In light of the above, we believe TVP makes significant contributions as a core advance in visual programming, backed by strong and comprehensive empirical evidence.

We deeply appreciate the reviewers' constructive feedback and the Area Chairs' efforts in evaluating our work.

---

### Meta-Review · Area_Chair_FkcY · 2026-01-06

**Summary:**

Three core concerns are summarized as follows:
- Evaluation Transparency Deficits: Unclear criteria for Example Library admission and tool abstraction potential assessment; undefined cyclomatic complexity computation; ambiguous threshold rationales affecting reproducibility.
- Generalization Limitations: Evaluated only on 3D spatial reasoning (no 2D benchmarks like MME/MMMU); reliance on proprietary GPT-4o, with unproven robustness on smaller/open-source models; initial doubts about genuine out-of-distribution generalization. Reviewers also raised questions regarding the impact of data ordering on the system and sought clarification on the mechanism for final answer selection.
- Efficiency Problem: Lack of initial quantitative cost/runtime data; multiple iterative LLM calls raise "meta-cost" concerns; modest performance gains vs. high computational overhead. Reviewers challenged the significance of the performance gains, noting that the improvements appeared marginal relative to the method's complexity.

**Reviewer Concerns:**

The reviewer's concerns may be addressed by the rebuttal:
- For the efficiency problem, the authors provided a detailed analysis of computational costs, estimating approximately $80 per iteration, demonstrating that the training and inference costs are manageable.
- For the generalization limitations, the authors conducted new experiments using the open-source Qwen2.5-Coder-32B, achieving an accuracy of 30.7%, which is comparable to the GPT-4o variant and outperforms the VADAR baseline. Through ablation studies, the authors demonstrated that the system exhibits better growth with random data ordering compared to a curriculum-based approach. They also clarified that the final answer does not rely on Ground Truth but is selected based on the highest score from the VLM judge for successfully executed programs.

The reviewer's concerns may still be outstanding:
- For the efficiency problem, the performance improvement remains marginal. Although the authors explained a 6.7% gain on the hard subset, the overall lift is limited, making the necessity of such a complex training pipeline debatable.

- For the generalization limitations, while the authors argued that 2D scenarios require only perception and reasoning without complex programming logic, 2D benchmarks remain critical. The lack of experiments verifying the framework's generalization capabilities on other visual reasoning tasks (beyond the tested 3D scenarios) is a limitation.

- For the evaluation transparency deficits, the evaluation mechanism relies on LLM-based judging with empirically selected hyperparameters, lacking validation or correlation checks against human experts.

**Reviewer Scores:**

- Reviewer uXe8/JN1p: Likely retain "marginally below acceptance" (residual concerns on LLM-judge reliability/incremental contribution).
- Reviewer fsyr: Maintain "marginally above acceptance" (core concerns addressed).
- Reviewer 44Xv: May raise score (all concerns resolved).

---

### Decision · Program_Chairs · 2026-01-26

Accept (Poster)